# Neighborhood characteristics and HIV treatment outcomes: A scoping review

**Linda Jepkoech Kimaru**[1]*, **Magdiel A. Habila**[2], **Namoonga M. Mantina**[1], **Purnima Madhivanan**[1], **Elizabeth Connick**[3], **Kacey Ernst**[2], **John Ehiri**[1]

**1** Department of Health Promotion Sciences, The University of Arizona, Tucson, Arizona, United States of America, **2** Department of Epidemiology and Biostatistics, The University of Arizona, Tucson, Arizona, United States of America, **3** Department of Medicine, The University of Arizona, Tucson, Arizona, United States of America

* lkimaru@arizona.edu

**Data Availability Statement:** Since its a scoping review, all articles are already available through the journal databases.

## Abstract

Recognizing challenges faced by people living with HIV is vital for improving their HIV treatment outcomes. While individual-level interventions play a crucial role, community factors can shape the impact of individual interventions on treatment outcomes. Understanding neighborhood characteristics' association with HIV treatment outcomes is crucial for optimizing effectiveness. This review aims to summarize the research scope on the association between neighborhood characteristics and HIV treatment outcomes. The databases PubMed, CINAHL (EBSCOhost), Embase (Elsevier), and PsychINFO (EBSCOhost) were searched from the start of each database to Nov 21, 2022. Screening was performed by three independent reviewers. Full-text publications of all study design meeting inclusion criteria were included in the review. There were no language or geographical limitations. Conference proceedings, abstract only, and opinion reports were excluded from the review. The search yielded 7,822 publications, 35 of which met the criteria for inclusion in the review. Studies assessed the relationship between neighborhood-level disadvantage (n = 24), composition and interaction (n = 17), social-economic status (n = 18), deprivation (n = 16), disorder (n = 8), and rural-urban status (n = 7) and HIV treatment outcomes. The relationship between all neighborhood characteristics and HIV treatment outcomes was not consistent across studies. Only 7 studies found deprivation had a negative association with HIV treatment outcomes; 6 found that areas with specific racial/ethnic densities were associated with poor HIV treatment outcomes, and 5 showed that disorder was associated with poor HIV treatment outcomes. Three studies showed that rural residence was associated with improved HIV treatment outcomes. There were inconsistent findings regarding the association between neighborhood characteristics and HIV treatment outcomes. While the impact of neighborhood characteristics on disease outcomes is highly recognized, there is a paucity of standardized definitions and metrics for community characteristics to support a robust assessment of this hypothesis. Comparative studies that define and assess how specific neighborhood indicators independently or jointly affect HIV treatment outcomes are highly needed.

**Funding:** The authors received no specific funding for this work.

**Competing interests:** The authors have declared that no competing interests exist.

## Introduction

The HIV pandemic has caused significant mortality, with 40.4 million AIDS-related deaths and an estimated 39 million people living with HIV globally in 2022 [1]. Antiretroviral therapies (ARTs) have improved health, and life expectancy, and reduced transmission [2–5]. Despite progress, disparities in infection risk and treatment access persist [6]. The UNAIDS 90-90-90 strategy aimed for 90% of PLWH to know their status, of those 90% receiving ART, and of those 90% achieving viral suppression, but targets were not fully met. As of 2022, only 76% of PLWH were accessing treatment, and of those 71% achieved viral suppression [1]. Understanding barriers to sustained treatment is vital to achieving the extended UNAIDS target of 95-95-95 by 2030 [7].

HIV treatment outcomes or HIV continuum performance, encompassing ART initiation, ART adherence, and HIV viral suppression [8], is influenced by both individual and community-level factors. Barriers such as poverty, limited resources, stigma, transportation and housing insecurity, food insecurity, discrimination, poor mental health, lack of support, and caregiving responsibilities hinder optimal HIV treatment outcomes [6, 9–12]. While studies focus on individual behavioral change in combating the barriers, the significance of community-level factors in sustaining behavior changes is often overlooked [13–15]. The Social Ecological Model (SEM) postulates that health is impacted by our physical and sociocultural environments [16, 17]. Recognizing these community-level influences is crucial for targeted interventions that can optimize the benefits of ART for people living with HIV.

The review's conceptual model (**S1 File**) is based on a five-level SEM framework assessing how an individual's access and utilization of HIV treatment and care is shaped by interactions with physical and sociocultural environments [18–20]. The conceptual framework (**S1 File**) illustrates our focus on community aspects like culture, resources, capacity, and the social and physical environment, all influencing HIV care access and utilization. All levels of the model are vital for sustained access and support for HIV treatment. Utilizing the conceptual model (**S1 File**), our scoping review delves deeper into the community level, emphasizing the intricate role neighborhoods play in shaping individual HIV treatment outcomes. The community level examines settings like neighborhoods, schools, and workplaces where social relationships take place, aiming to identify characteristics of these settings that are linked to health [21]. We focus specifically on the characteristics of neighborhoods in this review, defined as areas where people live and interact, often sharing similar incomes and social traits like education level, housing preferences, sense of public order etc. [22]. The physical and social attributes of a neighborhood have been recognized to influence health [23]. At the community-level pathways such as resource accessibility, behavioral norms, social support, and information networks, shape the context in which health-related decisions are made at the individual level [24–26].

Neighborhood characteristics play a crucial role in shaping various aspects of HIV treatment and care. Research has shown associations between neighborhood characteristics and HIV risk behaviors, transmission patterns, testing rates, and engagement in HIV treatment and care [27–35]. For instance, certain neighborhood factors, such as high rates of poverty, limited healthcare resources, or a lack of social support, can contribute to increased HIV risk behaviors and transmission [27–35]. In contrast, neighborhoods with better access to healthcare services, supportive social environments, and reduced stigma may facilitate higher rates of HIV testing and improved engagement in HIV treatment and care [27–35]. By understanding and examining neighborhood characteristics, we can gain insights into the contextual factors that influence individuals' behaviors and decisions regarding HIV prevention, testing, and treatment. This knowledge is essential for developing targeted interventions and strategies to

optimize the HIV continuum and improve health outcomes for individuals and communities affected by HIV.

The literature describes neighborhood characteristics in many ways using the tools, indicators, and datasets to characterize a neighborhood's socioeconomic status (SES) [36], deprivation index [37], disorder status [38], economic disadvantage [39], composition and interaction [40]. **Table 1** provides detailed definitions of these concepts and how they may impact health. Existing reviews assessing the role of neighborhood characteristics on HIV rarely separate HIV treatment outcomes (ART initiation, ART adherence, HIV viral suppression, etc.) from HIV prevention (HIV testing, condom use, sexual risk behavior, etc.) outcomes [41–44], making it difficult to identify points of community-level interventions for PLWH. Furthermore, current reviews focus more on neighborhood characterization of geographic boundaries and structural barriers, and less on the social characteristics of a neighborhood [41, 44, 45]. This review will address this gap by using a conceptual model guided by SEM [16, 17] as shown in **S1 File**. to understand how an individual is influenced by their physical, socio-economic, and cultural environments. The goal of this review is to examine the scope, nature, and extent of available research on the association between neighborhood characteristics and HIV treatment outcomes. It is hoped that the findings will inform efforts to integrate considerations for the role of neighborhood characteristics into HIV care.

## Methods

This scoping review was conducted in accordance with Arksey and Levac's methodological framework for scoping reviews which involves identifying a research process, selecting relevant studies, study selection, charting data, summarizing and reporting results [59, 60]. An iterative process of exchange, discussion, and literature review led to the development of the review's guiding question. The review's research question is 'What is our current knowledge on the impact of neighborhood characteristics on HIV treatment outcomes among adults?'. The question is centered on the conceptual model for the review (**S1 File**), exploring the community-level impacts of health. The research question is broad, explorative and aims for a comprehensive understanding of the research available on this topic, which leads us to the scoping review methodology. Neighborhood characteristics have been defined and measured in various ways and thus the flexibility of a scoping review in the study inclusion was ideal as opposed to strict inclusion criteria of a systematic review. Our findings are reported in agreement with the Preferred Reporting Items for Systematic Reviews and Meta-Analysis extension for Scoping Reviews (PRISMA-ScR) checklist [61]. This scoping review's protocol was registered on the Open Science Framework (OSF) website (DOI: 10.17605/OSF.IO/MD89T).

### Identifying relevant studies

The following databases were searched from the start of each database to November, 21, 2022: PubMed (MEDLINE), Embase (Elsevier), CINAHL (EBSCOhost), and PsycINFO (EBSCO-host). The start of each database as determined by the earliest publication. The earliest publication from each database is as follows PubMed (1966), Embase (1947), CINHAL (1937), and PsyhINFO (1887). A Medical Librarian (JM) developed a search strategy for PubMed using the following terms and concepts: (Neighborhood OR Community-level OR "neighborhood characteristics" OR "neighborhood environment" OR "neighborhood socioeconomic status" OR "neighborhood deprivation" OR "neighborhood disadvantage" OR "residential characteristics" OR "neighborhood disorder" OR "physical decay" OR "physical disorder" OR "social disorder") AND (HIV OR "Human immunodeficiency virus" OR "HIV treatment" OR "HIV viral suppression" OR "antiretroviral therapy" OR "HIV viral load" OR "antiretroviral therapy

**Table 1. Neighborhood characteristics definitions.**

| | |
|---|---|
| **Neighborhood Composition and interaction** | Referring to the socio-demographic makeup of the neighborhood and its residents, including relationships and social processes that exist among people living in the neighborhood [40]. This includes and is measured with items such as racial/ethnic make-up, age, gender, collective efficacy, social cohesion, and social capital. The racial/ethnic density effect theory posits that racial/ethnic individuals living in areas concentrated with people of the same racial/ethnic group are healthier than those in areas of less concentration [46]. Social support networks provide emotional and practical assistance, bolstering mental well-being and treatment adherence [47]. High levels of social cohesion foster community engagement, trust, and shared values, potentially enhancing stress management and health behaviors [48]. Collective efficacy, the belief in collective problem-solving, can lead to safer environments and improved well-being through collaborative health initiatives [49]. Social capital's resources and relationships offer access to health-enhancing information and networks [50]. |
| **Neighborhood Socio-economic Status (SES)** | Referring to the overall marker of neighborhood conditions that may define residents' access to health care independent of their individual characteristics [36]. In line with the definition, this included and is measured by neighborhood level, residential stability, education, income, employment, car ownership, house ownership, type of household, healthcare access, food access, etc. A neighborhood's SES describes the social standing or class of its residents and is often measured as but not limited to a combination of education, income, and occupation [51]. SES is an important determinant of access to healthcare as persons with low SES compared to those in high SES are more likely to be uninsured, have worse self-reported health, have lower life expectancies, suffer from more chronic conditions, have poor-quality healthcare, and seek health care less often; when they do seek health care, it is more likely to be for an emergency [52, 53]. |
| **Neighborhood Deprivation** | Referring to and measured by the relatively low physical (e.g., abandoned homes, graffiti, etc.), social (e.g., loitering, unemployment, etc.) and economical position (e.g., education, income etc.) of a neighborhood [37]. Neighborhood deprivation is commonly characterized by indicators such as unemployment, low income, low education and low-paying jobs, residential vacancy, graffiti, etc. [37, 54]. Living in highly deprived neighborhoods has been associated with risky health behaviors [55]. |
| **Neighborhood Disorder** | Referring to observed or perceived physical (e.g., trash, vandalism, etc.) and social (e.g., over policing, homelessness etc.) features of neighborhoods that may signal a breakdown of order and social control that can undermine the quality of life [38]. In line with the definition, we regarded and measured social disorder, physical disorder, policing, crime rates, incarceration rates, and perceived fear as components of neighborhood disorder. A meta-analysis found that perceived neighborhood disorder was consistently associated with mental health outcomes, as well as substance abuse, and measures of overall health [26]. |
| **Neighborhood Disadvantage** | Referring to and measured as a community or neighborhood where the percentage of households below the poverty line is greater than a critical prevalence [39]. Disadvantaged neighborhoods affect health by limiting access to proper nutrition, shelter, safe neighborhoods, clean air and clean water, utilities, and other items that define an individual's standard of living [39, 56]. |
| **Rural/urban status** | The census bureau defines and measures rural areas as comprising open country and settlements with fewer than 2,500 residents, while urban areas are defined as any place that has an urban nucleus of 50,000 or more people, a core with a population density of 1,000 persons per square mile, and may contain adjoining territory with at least 500 persons per square mile [57]. Census units are important for public health policy and resource allocation. Census data provides a wealth of information on the population and is used to ensure equal representation and access to important governmental and private sector resources [58]. The information also guides allocation of monies for community decision-making that impacts schools, housing, health care services, employment opportunities, etc. [58]. |

adherence" OR "antiretroviral therapy initiation" OR "ART initiation" OR "ART adherence").
These terms were adapted for other three databases. The search was not limited by language,
geography, or date of publication. We used general search terms to capture indicators that are
commonly used in the literature to describe neighborhood characteristics. However, indicators
that were less common such as (eg. physical decay, physical or social disorder) we included
those terms explicitly as part of the search strategy.

Results from the search were imported into EndNote 10X (Clarivate), and duplicates were
removed using EndNote's duplicate identification strategy. Deduplicated studies were
uploaded into Rayyan QCRI for screening, it is a free web-based tool designed for systematic
reviews or other synthesis projects [62]. Publication screening was conducted in two phases:
title and abstract screening followed by full text review for eligibility. Each publication from
the search was screened independently by two reviewers (LJK and MH or LJK and NM). Dis-
agreements between two reviewers were resolved by a third reviewer (MH or NM) who did
not participate in the previous screening phase.

**Eligibility criteria.** All experimental designs (pre-experimental, quasi-experimental, and
true-experimental) and observational study designs (cross-sectional, case report or case series,
case-control, and cohort studies) were eligible for inclusion. The study population of interest
was adults 18 years old and above who were living with HIV. We included all studies that
examined the association between neighborhood characteristics and HIV treatment outcomes.
Neighborhood characteristics were defined as fitting to or resembling at least one of the follow-
ing definitions presented in **Table 1.** The HIV treatment outcomes for this review were
defined as having at least one of the following indicators: i) Antiretroviral Therapy (ART) initi-
ation (the start of an HIV treatment plan, measured by enrollment into ART [63]); ii) ART
adherence (an individual's ability to follow an ART treatment plan, take medications at pre-
scribed times and frequencies, and follow relevant restrictions [64]); and iii) HIV viral sup-
pression (reduction of HIV viral load to very low levels [65]). Viral load is recommended as
the preferred monitoring approach to confirm treatment failure [66]. Exclusion criteria
included studies that did not pertain to neighborhood characteristics and HIV treatment out-
comes, studies where only an abstract was available, and studies that reported opinions.

**Data charting and reporting.** We abstracted the data from the publications in three cate-
gories: i) publication and study information (authors, year of publication, study objective,
study design, data source, sample size, study population, and study setting), ii) exposure infor-
mation (definition and measures used by the study authors for neighborhood characteristics),
iii) outcome information (ART initiation, ART adherence, and HIV viral suppression). For
neighborhood characteristics, we abstracted data that described the association between its
indicators (detailed in **Table 1**) and an HIV treatment outcome. This is to highlight which spe-
cific neighborhood indicator may have an impact on HIV treatment outcomes. Data abstrac-
tion for each included study was completed by two reviewers (LJK and MH or LJK and NM).
A narrative synthesis approach was used to report results. Quality Assessment of included
studies was not performed as it is unnecessary for scoping review methodology and was not
necessary to meet the study objectives [60].

## Results

### Study characteristics

The systematic search yielded 5,211 unique entries after removal of duplicates as seen in **Fig 1**.
After screening, 35 studies with a total of 277,856 participants were included in the review
[67–101]. In **Table 2**, the majority of the studies (n = 28, 80%) were published between 2016
and 2022. Most studies were retrospective secondary analysis (n = 16, 46%), in high income

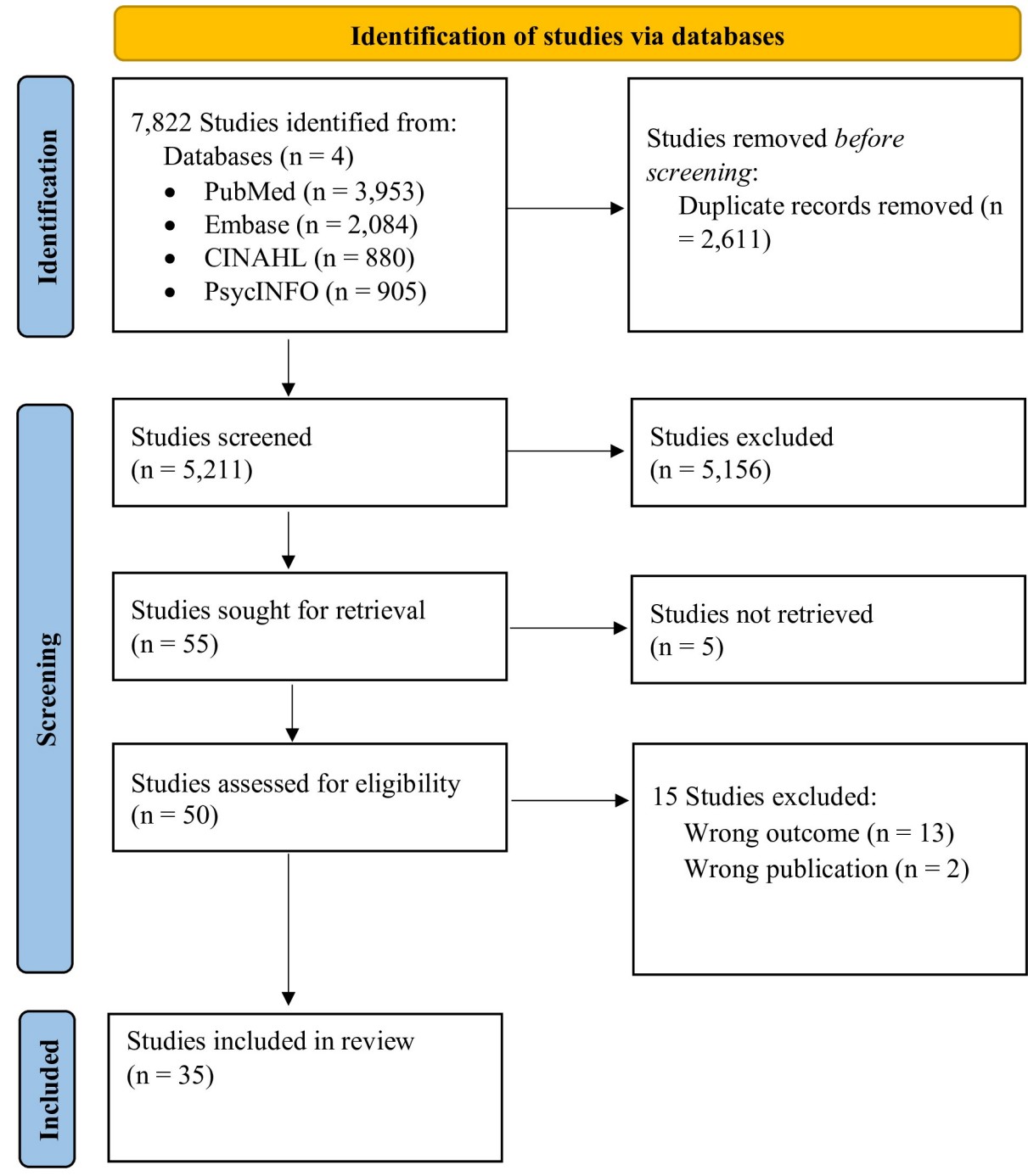

**Fig 1. PRISMA flow diagram for the screening process.**

countries (n = 34, 97%), and urban settings (n = 25, 71%) in the United States. Almost all the studies (n = 33, 94%) used census data to the characterize neighborhoods. The studies also used various characteristics to define neighborhoods such as disadvantage (n = 25, 71%), SES (n = 23, 66%), composition and interaction (n = 18, 51%), deprivation (n = 17, 49%), and disorder (n = 11, 31%). In **Table 3**, we provide the study description of each of the studies and the reported relationships between neighborhood characteristics and HIV treatment outcomes.

**Table 2. Descriptive statistics of included studies.**

|  | Total (N = 35) n (%) |
|---|---|
| **Year of Publication** |  |
| 2006–2010 | 4 (11%) |
| 2011–2015 | 3 (9%) |
| 2016–2020 | 22 (63%) |
| 2021 –Nov 2022 | 6 (17%) |
| **Study Design** |  |
| Cross-sectional | 7 (20%) |
| Cohort Study | 11 (31%) |
| Retrospective secondary analysis | 16 (46%) |
| Case-control study | 1 (3%) |
| **Country** |  |
| United States | 30 (86%) |
| Canada | 2 (6%) |
| Switzerland | 1 (3%) |
| South Africa | 1 (3%) |
| Brazil | 1 (3%) |
| **Urban/rural setting** |  |
| Urban | 25 (71%) |
| Both urban and rural | 10 (29%) |
| **Sample size** |  |
| <1000 | 12 (34%) |
| 1000–5000 | 13 (37%) |
| 5001–10,000 | 4 (11%) |
| 10,001–15,000 | 1 (3%) |
| 20,001–25,000 | 1 (3%) |
| 25,001–30,000 | 2 (6%) |
| >30,000 | 2 (6%) |
| **Neighborhood Units** |  |
| Census tract | 13 (37%) |
| Census Zip Code Tabulation Areas (ZCTA) | 9 (26%) |
| Census Zip Codes | 6 (17%) |
| Census geocodes | 2 (6%) |
| Census blocks | 2 (6%) |
| None | 2 (6%) |
| Census Rural-Urban Commuting Areas (RUCA) only | 1 (3%) |
| **Characteristics (not mutually exclusive)** |  |
| Neighborhood disadvantage | 25 (71%) |
| Neighborhood composition and interaction | 18 (51%) |
| Neighborhood socioeconomic status (SES) | 23 (66%) |
| Neighborhood deprivation | 17 49%) |
| Neighborhood disorder | 11 (31%) |

**Neighborhood socioeconomic status (SES).**   This construct focuses on the range of economic and social resources of a neighborhood's residents. Among the 23 studies that discussed neighborhood level SES, two studies discussed its outcomes on ART initiation [67, 79], two studies discussed ART adherence [81, 86], 15 studies discussed HIV viral suppression[70, 72, 77, 78, 80–82, 88, 91, 93–95, 99, 100], and 6 studies combined neighborhood SES factors into a

**Table 3. Studies description and reported relationships between neighborhood characteristics on HIV treatment outcomes.**

| No. | Study | Population | Exposure: Neighborhood characteristics | HIV treatment outcomes |
|---|---|---|---|---|
| | Joy et al. 2008 | Men and women, aged 18 years and older in the HIV/AIDS drug treatment program, British Columbia, Canada | Six variables to be used as measures of socioeconomic status at the census tract-level were obtained from the census data:<br>• Percent of residents with post-secondary education<br>• Percent of unemployed residents<br>• Percent of aboriginal residents<br>• Percent of residents living below the Canadian poverty line<br>• Median neighborhood income<br>Additionally, whether the neighborhood was urban or rural. | **ART initiation:** Defined by a CD4 count of <50 cells/mm3.<br>• Percent of residents with post-secondary education, percent of aboriginal residents, percent of residents living below the Canadian poverty line, median neighborhood income, and rural/urban status were not associated with late access to HAART<br>• Unemployment was associated with delayed access to treatment (odds ratio = 1.41, 95% confidence interval [CI]: 1.14 to 1.74) when controlling for age, previous AIDS-defining illness, and baseline plasma viral load. |
| | Arnold et al. 2009 | Men and women, aged 15 years and older in the San Francisco HIV surveillance Registry between 1996 to 2000 and followed through 2006, United States | Neighborhood SES context (NSEC) indicators at the census tract-level considered in construction of the summary measure included:<br>• Proportion of residents by race/ethnicity<br>• Proportion of residents by annual household income level<br>• Log per capita income<br>• Proportion of residents by age range<br>• Proportion male<br>• Proportion of males in the labor force<br>• Proportion of employed males in the labor force<br>• Proportion of each gender group in specified education levels<br>• Log median household value in US dollars<br>The final variables were converted to z-scores, weighted by the factor loadings, and summed to determine final scores. The final NSEC scores ranged from -17 to 14. Quartile ranges were used to identify four NSEC levels: low (-17 to -5), moderate-low (-5 to -0.1), moderate-high (0 to 4.9) and high (5 or greater). | **ART initiation:** Defined as delays in treatment were estimated using the timing of ART initiation. When used as an explanatory variable in the survival models ART initiation was grouped into five categories: (i) more than one year prior to diagnosis, (ii) within one year prior to diagnosis, (iii) between 0 and 59 days after diagnosis, (iv) 60 days or more after diagnosis, and (v) never.<br>• Those residing in higher NSEC neighborhoods were less likely to have delayed or no ART initiation relative to residents of lower NSEC neighborhoods. Put differently, there was 40% greater odds of no treatment if one resided in the lowest NSEC compared to residents in the highest NSEC. |
| | Druyts et al. 2009 | Men and women, aged 18 years and older in the HIV/AIDS drug treatment program, British Columbia, Canada | The census tract-level socioeconomic variables included:<br>• Percentage of residents with post-secondary education,<br>• Percentage of residents who were unemployed,<br>• Percentage of residents who were Aboriginal,<br>• Percentage of residents living below the Canadian poverty line<br>• Median neighborhood income<br>Initial analyses indicated that the census-based variables were highly correlated, thus not included in the multivariable analysis. Instead, the following neighborhood variables were used instead<br>• High concentration of injecting drug users<br>• High concentration of gay men | **ART initiation:** Defined as the number of days for which medication was dispensed divided by the number of days for which medication was prescribed in the first year of treatment. Adherence measures were dichotomized as <95% (nonadherent) and ≥95% (adherent).<br>• Of the 533 patients who initiated HAART, 287 (54%) lived in the neighborhood with a high concentration of gay men, and 246 (46%) lived in the neighborhood with a high concentration of injecting drug users.<br>• Compared with patients prescribed highly active antiretroviral therapy (HAART) residing in the neighborhood with a high concentration of gay men, patients residing in the neighborhood with a high concentration of injecting drug users were assigned to census tracts with significantly lower mean levels of:<br>○ Percentage with post-secondary education [median (IQR)] 33.6 (31.3–35.4) versus 11.4 (9.3–11.4), with a p<0.001<br>○ Percentage Aboriginal [median (IQR)] 2.0 (1.8–2.4) versus 14.8 (10.0–14.8), with a p<0.001<br>○ Percentage with low income [median (IQR)] 30.2 (25.9–30.4) versus 71.0 (71.0–80.0), with a p<0.001<br>○ Median income (CAN $) [median (IQR)] 25835 (25338–29017) versus 9886 (9886–10204), with a p<0.001<br>○ Unemployment rate [median (IQR)] 7.3 (7.1–7.9) versus 24.2 (24.2–25.0), with a p<0.001 |
| | Ohl et al. 2010 | Men and women in the Veterans Aging Cohort study in Iowa City, United States | Geographical Units–Census with unspecified unit<br>• Rural residence was determined by linking ZIP codes to Rural Urban Commuting Area (RUCA) codes. RUCA codes are measures of rurality used for research and policy that incorporate population density as well as urban commuting patterns.19,20 The 33 RUCA codes were collapsed into 2 categories, rural and urban, using a standard algorithm.19 | **ART initiation:** cART initiation, which we defined as a minimum of 3 antiretroviral medications, and engagement in HIV primary care as reflected by outpatient infectious disease and general medicine clinic visits.<br>• Rural persons were somewhat more likely to initiate cART in total (74.9% vs. 68.7%, P = 0.001) |
| | Shacham et al. 2013 | Men and women attended the Washington University HIV Clinic in St Louis, MO in 2008 | To characterize neighborhoods, we analyzed three variables at the census-tract level:<br>• Percentage poverty;<br>• Percentage of the census tract that was majority African American<br>• Percentage of unemployment | **ART Initiation:** Defined as the use of at least three drugs from two different antiretroviral drug classes or the use of at least three nucleoside reverse transcriptase inhibitors (NRTIs).<br>After adjusting for age, gender, race, education, income, depression (except depression outcome), number of sex partners and unprotected sex events.<br>• Percent poverty and percentage of the census tract that was majority African American was not associated with a current ART prescription with an OR of 0.70 (95% CI: 0.47–1.05) and OR of 0.79 (95% CI: 0.52–1.18) respectively.<br>• In neighborhoods with higher rates of unemployment, individuals were less likely to have a current ART prescription (OR 1.47; 95% CI 1.05–2.04).<br>**HIV Viral suppression:** Virological suppression was considered <400 HIV-1 RNA copies/mL at the time of these analyses.<br>• Percentage poverty (OR 0.78; 95% CI 0.55–1.12), percentage of the census tract that was majority African American (OR 0.79; 95% CI 0.55–1.14) and percentage of unemployment (OR 0.84; 95% CI 0.59–1.21) were not associated with HIV viral load. |

*(Continued)*

**Table 3.** (Continued)

| No. | Study | Population | Exposure: Neighborhood characteristics | HIV treatment outcomes |
|---|---|---|---|---|
| | Gueler et al. 2015 | Men and women 16 years and older, who were treatment-naive and started cART between 2000 and 2013 in the Swiss HIV Cohort Study, Switzerland | The social economic position (SEP) of neighborhoods was determined by:<br>• rent per square meter<br>• proportion of households headed by a person with primary education or less,<br>• proportion headed by a person in manual or unskilled occupation<br>• mean number of persons per room. | **HIV viral suppression:** Virologic response to cART was defined as a viral load less than 50 copies/ml at 6 months among ART-naive patients followed for at least 6 months.<br>• Patients living in neighborhoods in the highest quintile of SEP were more likely to suppress viral replication at 6 months after starting cART than those living in neighborhoods from the lowest quintile: OR 1.52 (95% CI 1.14–2.04, p<0.05). |
| | Surratt et al. 2015 | Men and women, 18 years old and above who are HIV positive, active substance users, and current prescription for ARV medication in all major HIV service organizations in North and Downtown Miami, United States | • Using a perceived neighborhood disorder scale, a standardized 10-item scale that captures elements of social and physical disorder in the neighborhood environment. Scores ranged from 8 to 40, with higher scores indicating greater perceived neighborhood disorder.<br>• Neighborhood poverty level was examined using residential zip codes reported by study participants. We categorized zip codes by percentage of individuals below the poverty level. | **ART adherence:** We used total ARV doses prescribed and total doses missed in this 7-day period to generate an adherence percentage score.<br>• Participants reporting higher neighborhood disorder had higher odds of diversion-related nonadherence OR 1.03 (95% CI 1.01, 1.05)<br>• Percentage of individuals below the poverty line had lower odds of diversion-related nonadherence OR 0.98 (95% CI 0.96, 0.99) |
| | Burke-Miller et al. 2016 | Women in the Women's Interagency HIV Study (WIHS) with a median age of 47 in Chicago, United States | Perceptions of Neighborhood Environment Scale (PNES):<br>• built environment (e.g., trash and litter; poor building and sidewalk maintenance; lack of shade trees; little opportunity for walking and outdoor exercise use; heavy traffic)<br>• food desert<br>• safety and violence<br>• social cohesion.<br>These estimates were dichotomized at the median to create high/low measures of each domain of neighborhood disorder.<br>We also used Community area (CA) census measures: of<br>• concentrated poverty (≥25% of residents below federal poverty level)<br>• racial segregation (≥50% Black non-Hispanic residents). | **HIV Viral suppression:** non-suppressed viral load was defined as ≥200 copies/ml versus lower or undetectable.<br>Adjusting for individual characteristics, there were no statistically significant or marginal associations between neighborhood characteristics and non-suppressed viral load.<br>• Adjusting for White, Hispanic, age, stable housing, high school or more education, income <$18k, IDU ever, Chicago (versus suburb), and cART non-use or nonadherence.<br>○ Poor quality environment, in unadjusted analysis had an OR 1.43 (95% CI: 0.68, 2.98; p = 0.342) and in adjusted analysis had an OR of 1.06 (95% CI: 0.39, 2.90; p = 0.910)<br>○ Food desert, in unadjusted analysis had an OR of 1.74 (95% CI: 0.84, 3.62; p = 0.138) and in adjusted analysis had an OR of 0.99 (95% CI: 0.37, 2.71; p = 0.992)<br>○ Unsafe environment, in unadjusted analysis had an OR of 2.13 (95% CI: 0.94, 4.81; p = 0.069) and in adjusted analysis had an OR of 1.45 (95% CI: 0.51, 4.16; p = 0.484)<br>○ Low cohesion, in unadjusted analysis had an OR of 1.85 (95% CI: 0.86, 3.98; p = 0.116) and in adjusted analysis an OR of 1.08 (95% CI: 0.41, 2.87; p = 0.874)<br>Concentrated poverty ≥25% below the federal poverty line, in unadjusted analysis had an OR of 2.58 (95% CI: 1.27, 5.25; p = 0.010), in adjusted analysis had an OR of 2.19 (95% CI: 0.71, 6.74; p = 0.169)<br>○ Racial segregation ≥50% Black non-Hispanic, in unadjusted analysis had an OR of 2.70 (95% CI: 1.28, 5.70; p = 0.009), in adjusted analysis had an OR of 1.08 (95% CI: 0.41, 2.87; p = 0.869) |
| | Richardson et al. 2016 | Male and Female veterans in the Veterans Health Association Corporate Database, United States | The created a principle components factor score that combined five variables measured at the census tract level:<br>• percentage of persons that lived below the poverty line<br>• percentage of persons age 16 and older employed in professional and managerial occupations<br>• percentage of households that were female-headed,<br>• percentage of males aged 16 and older who were unemployed or not in the labor force; and<br>• median household income.<br>The principle components score was used to create a 5-level categorical measure of neighborhood social disadvantage. The top 20% of scores are classified as very high disadvantage, followed by the next 20% which are labeled high disadvantage, then moderate, low, and very low disadvantage. | **ART initiation:** Receipt of cART was defined as medication fills in 2013 from 2 or more classes of antiretroviral drugs<br>**HIV Viral suppression:** viral control was defined as HIV serum RNA less than 200 copies per milliliter<br>In unadjusted analyses, blacks were less likely than whites to receive cART or experience viral control. The differences persisted in multivariable models adjusting for patient characteristics and neighborhood social disadvantage: blacks had lower odds than whites of receiving cART or experiencing viral control.<br>• The relationship between race and HIV viral control was attenuated by adjustment for patient characteristics and neighborhood social disadvantage, with an unadjusted odds ratio of 0.53 (95% CI 0.48–0.57) and fully adjusted odds ratio of 0.68 (95% CI 0.62–0.75). |
| | Shacham et al. 2017 | Men and women, 18 years and above attending the Washington University HIV Clinic, United States | Perceived Neighborhood Disorder Scale ranges from 1 to 4 (strongly agree to strongly disagree) about their perceived level of danger in their residence.<br>Collective Efficacy Scale—range from 1 to 5 (very likely to very unlikely), assessing the social cohesion, specifically how likely are you to rely on neighbors to address concerns in the neighborhood. Some examples include, neighbors being willing to help each other, feeling trust among the people in the neighborhood, and likelihood of neighborhood doing something to reduce spray graffiti on local buildings.<br>The Perceived Fear Scale- measures how many total days participants had feared personal safety, home safety, and neighborhood safety within the past 7 days. The range was from 0 to 7. | **ART initiation:** defined as the use of 3 antiretroviral drugs.<br>**HIV viral suppression:** HIV viral loads were used as a proxy for medication adherence and HIV viral loads were dichotomized as follows (<400 copies/mL and ≥400 copies/mL)<br>There were no significant associations between sociodemographic characteristics and HIV management characteristics (current HIV viral loads, and current receipt of cART prescription) with the social support and the collective efficacy scale. Reports of social support and neighborhood perceptions were highly correlated. Greater levels of perceived fear were independently associated with having current cART prescriptions (p = 0.008) |

*(Continued)*

**Table 3.** (Continued)

| No. | Study | Population | Exposure: Neighborhood characteristics | HIV treatment outcomes |
|---|---|---|---|---|
| | Wiewel et al. 2017 | Men and women, aged 13 years and older in the New York City (NYC) HIV Surveillance Registry between 2006 and 2010, United States | The two neighborhood SES measures were used were percent of residents with incomes under the federal poverty threshold and percent unemployed among residents aged 16 years and older.<br>• Neighborhood poverty was categorized as 0 to <10% (low poverty), 10 to <20% (medium poverty), 20 to <30% (high poverty), and 30–100% (very high poverty).<br>• The values for percent unemployed and percent Black were divided by 10 to estimate hazard ratios for 10-percentage point differences. | **HIV viral suppression:** virologic failure, defined as having a VL >1000 copies/mL.<br>• The proportion of persons achieving viral suppression within 12 months of diagnosis increased as neighborhood poverty decreased.<br>• When compared to persons who maintained suppression, those who experienced failure within 12 months of first suppression were most likely to have been residents of higher-poverty neighborhoods (e.g., 22.4% of residents in very-high-poverty neighborhoods vs. 12.8% in low-poverty experienced failure, p<0.0001)<br>• Although marginally significant, each 10% increase in the proportions of neighborhood residents who were unemployed or Black was associated with a 5% (95% confidence interval [CI] = 0.92–0.99) or 1% (95% CI = 0.98–0.99) lower likelihood of suppression, respectively. After adjusting for individual characteristics alone, and then individual characteristics as well as other neighborhood characteristics, neighborhood characteristics were no longer associated with suppression.<br>• The unadjusted hazard ratios show higher failure rates among residents of neighborhoods with higher proportions of residents who were poor, unemployed, or Black. Only high or very high neighborhood poverty remained associated with failure in models adjusted for individual and other neighborhood characteristics (e.g., adjusted hazard ratio = 1.19 [95% CI = 1.06–1.34] for very high vs. low poverty). |
| | Goswami et al. 2016 | Men and women, 13 years old and above in the Atlanta AIDSVu CDC's national HIV surveillance database, United States | The place-based covariates for this analysis included ZCTA-level transportation variables, provider availability, educational attainment, income inequality, poverty level, residential vacancy, and alcohol outlet density.<br>• Educational attainment was defined as the percentage of people in a ZCTA over 25 years of age with at least a high school diploma or equivalent.<br>• Income inequality was measured by the Gini coefficient, where zero represents perfect equality and one represents maximal inequality<br>• Poverty was defined as the percent of the population in a ZCTA living below the national poverty line. ZCTAs with higher poverty rate than the average poverty rate in Georgia (18.5%, 2009–2013) were defined as high-poverty and ZCTAs with poverty rate below the average poverty rate were classified low-poverty.<br>• We used two variables as proxies for transportation vulnerability: public transit access and car ownership.<br>• Because the U.S. Census defines residential vacancy based on recently constructed, vacant homes in high SES neighborhoods in addition to abandoned housing, we used a modified variable for percentage of vacant houses in a ZCTA to better reflect only the latter<br>• We used a measure of alcohol outlets by ZCTA based on licensing information obtained from the Georgia alcohol board<br>To create the density of alcohol outlets, alcohol outlet count was divided by the total area in square miles for each ZCTA. The same method was used to create density of residential vacancies from absolute number of vacant houses. | **HIV Viral suppression:** The percentage virally suppressed was defined as the number of persons who achieved a viral load less than 200 copies/mL.<br>High poverty (p<0.001, R² = 0.16), high alcohol outlet density (p = 0.04, R² = 0.07), low car ownership (p<0.001, R² = 0.11), greater number of bus stops (p = 0.02, R² = 0.01), lower educational attainment (p = 0.002, R² = 0.07), and greater percentages of vacant housing in the area (p = 0.05, R² = 0.06) were significantly associated with low levels of community viral suppression.<br>High car ownership was significantly associated with suppression in the low-poverty stratum; every 10-percentage point increase in household vehicle ownership in a ZCTA was associated with a 30-percentage point increase in ZCTA-level viral suppression (p = 0.01, R² = 0.08).<br>High bus stop count was associated with viral suppression in the high-poverty stratum but with less impact; every additional 10 bus stops in a ZCTA was associated with a 0.08% increase in ZCTA-level viral suppression (p <0.01, R² = 0.20). |
| | Sheehan et al. 2017 | Males and Females, 13 years old in the Florida HIV surveillance registry between 2000–2014, United States | Seven neighborhood-level socioeconomic status (SES) indicators were selected for use:<br>• percent of population living below the poverty line,<br>• median household income in 2013,<br>• percent of households with annual income <$15,000,<br>percent of households with annual income ≥$150,000,<br>• income disparity (derived from percent of households with annual income <$10,000 and percent of households with annual income >$50,000),<br>• percent of population aged >25 with less than a 12th grade education,<br>• percent of population aged >16 years employed in high working-class occupation (ACS occupation group: managerial, business, science, and arts occupations).<br>All neighborhood-level indicators were coded so that higher scores corresponded with lower SES (higher disadvantage); they were then standardized.<br>Extracted the percent of the population who identified themselves as non-Latino black from the ACS to describe racial composition of neighborhoods<br>Categorized ZCTAs into rural or urban, using Categorization C of Version 2.0 of the rural–urban commuting area (RUCA) codes, developed by the University of Washington. | **HIV viral load suppression:** defined as having a viral load of <200 copies/mL.<br>No associations between neighborhood-level SES and HIV viral suppression were found.<br>Rural residence was protective for non-Latino Blacks (aOR = 0.65; 95% CI 0.56–0.75) and non-Latino Whites (aOR = 0.8; 95% CI 0.68–0.97). |

*(Continued)*

**Table 3.** (Continued)

| No. | Study | Population | Exposure: Neighborhood characteristics | HIV treatment outcomes |
|---|---|---|---|---|
| | Xia et al. 2017 | Alive men and women at least 18 years old or above by December 2014 in the New York HIV surveillance registry, United States. | Census tract percentage of census tract residents living below the federal poverty level category (<10%, 10 - <20%, 20 - <30%, and ≥30%). | **HIV Viral Suppression:** Defined as the last viral load measure with a value of <200 copies/mL. Blacks living in the most impoverished neighborhoods had the lowest proportion of viral suppression, with 75% in males and 76% in females.<br>The highest proportion of viral suppression was seen in Whites living in the least impoverished neighborhoods (92% males and 90% females suppressed).<br>Among females, there were literally no differences in viral suppression by race/ethnicity among those living in the most impoverished neighborhoods (Blacks: 76%, Hispanics: 78%, and Whites: 79%).<br>Among males, Compared with Blacks living in the most impoverished neighborhoods, Whites in the most impoverished neighborhoods and Blacks in the least impoverished neighborhoods were 11% (PR = 1.11; 95% CI: 1.08, 1.14) and 7% (PR = 1.07; 95% CI: 1.04, 1.11) more likely to be virally suppressed, respectively.<br>Within each race/ethnicity, for every level of decrease in census tract poverty, Black, Hispanic, and White males were 2%, 1%, and 1%, respectively, more likely to be virally suppressed.<br>Among females, Compared with Blacks living in the most impoverished neighborhoods, Blacks in the least impoverished neighborhoods, Whites in the most impoverished neighborhoods, and Whites in the least impoverished neighborhoods were 2% (PR = 1.02; 95% CI: 0.98, 1.06), 5% (PR = 1.05; 95% CI: 0.98, 1.12), and 14% (PR = 1.14; 95% CI: 1.09, 1.19) more likely to be virally suppressed, respectively.<br>Within each race/ ethnicity, for every level of decrease in census tract poverty, Black, Hispanic, and White females were 1%, 3%, and 3%, respectively, more likely to be virally suppressed. |
| | Jefferson et al. 2017 | Men who have sex with men (MSM), 13 years and above in the New York UHF registry between 2009–2013 living in a NYC UHF district, United States | Neighborhood SES was measured using demographic composition, economic disadvantage, and healthcare access at the UHF-level.<br>Demographic composition:<br>low (<30 of 10,000 households MSM-headed), medium (30 to <60), and higher (≥60) and low (<5% of residents were Black), medium (5 to 29%), and high (≥ 30%)<br>• The percent of residents who were 21–54 years old<br>Economic disadvantage:<br>• median income<br>• percent of people aged >16 years in the workforce who were unemployed<br>• percent of individuals at or below federal poverty level<br>• percent of households that had received public assistance in the last 12 months<br>• percent of adults ≥25 years old who did not have a high school degree/GED<br>Healthcare Access:<br>• percent of UHF residents without health insurance<br>• percent of UHF residents who reported an unmet need for care in the last 12 months<br>Residential vacancy<br>• the number of residential homes or apartments per square mile that had once been occupied but were now vacant<br>Social Disorder:<br>• Police Stop and Frisk—low (<6000 stops without arrest per 100,000 residents), medium (6000 to <22,000), and high (>22,000). | **HIV Viral suppression:** "virally suppressed" was defined as having ≤200 copies of HIV per ml of blood at any point within the 12 months after diagnosis; "durably virally suppressed" if, within 12 months after their diagnosis, they (A) had at least two suppressed viral load tests that were at least 90 days apart with no intervening unsuppressed viral load tests, and (B) had no unsuppressed viral load tests after they had achieved durable viral suppression<br>Demographic composition:<br>Bivariate and multivariable models suggest that Black MSM living in a UHF with a low concentration of MSM-coupled households (<30/10,000) were 18% less likely to achieve durable suppression than those living in a UHF with a higher concentration of MSM coupled households (<60/10,000; RR = 0.80, p = 0.01; ARR = 0.82, p = 0.05).<br>Among Black MSM, the chances of achieving durable suppression were also less if they lived in a UHF with a medium concentration of MSM-coupled households (30 to <60 MSM-coupled households/ 10,000 households) compared to a higher concentration (RR = 0.90, p = 0.14; ARR = 0.89, p = 0.10). Relationships between the concentration of MSM-coupled households and durable suppression were the same for Latino MSM as for Black MSM (i.e., the Latino interaction p value was not significant).<br>MSM living in UHF districts where 5–29% of residents were Black were more likely to be suppressed than MSM living in districts where >30% residents were Black [relative risk (RR) = 1.14, p<0.0001].<br>As with Black and Latino MSM, models indicate that White MSM had a lower likelihood of durable suppression if they lived in a UHF with a low concentration of MSM-coupled households (i.e. the White interaction p-value comparing low to high was not significant). However, for White MSM, as compared to Black MSM, living in a UHF district with a medium concentration of MSM-coupled households, compared to a UHF with a high concentration, was also associated with a larger chance of achieving durable suppression (RR = 1.22, p = 0.03; ARR = 1.26, p = 0.01).<br>MSM living in UHF districts where 5–29% of residents were Black were 7% more likely to be suppressed than MSM living in districts where <30% of residents were Black [adjusted relative risk (ARR) = 1.07, p = 0.04].<br>Economic disadvantage<br>Economic disadvantage did not have a significant association with viral suppression in the bivariate models (RR = 0.99 p = 0.23)<br>Healthcare access<br>Poor access to healthcare was not significant associated with viral suppression (RR = 1.01, p = 0.96).<br>Residential vacancy<br>Residential vacancy was significantly associated with viral suppression in bivariate models but not in multivariable models, (RR = 0.999, p = 0.01) and (RR = 0.9997, p = 0.53) respectively.<br>Social disorder<br>In both bivariate and multivariable models, there was no significant association between rates of stop without arrest and suppression for Black or Latino MSM. However, for White MSM living in UHFs with a medium rather than a high number of stops (i.e., between 6000 and 22,000 stops without arrest versus [22,000 stops without an arrest per 100,000 residents) was associated with lower viral suppression (RR = 0.84, p = 0.06; ARR = 0.84, p = 0.05). |
| | Rebeiro et al. 2018 | Men and women, 18 years old and older in attending the Vanderbilt Comprehensive Care Clinic, United States | For each ZCTA, the neighborhood socioeconomic contextual indicators included:<br>• percentage of the population of black race<br>• median age<br>• percentage with male sex assigned at birth (sex)<br>• percentage living below twice the Federal Poverty Level<br>• per capita income<br>• percentage with less than a high school education<br>• percentage not participating in the labor force<br>Z-scores for each indicator were calculated then summed across indicators for everyone to create a neighborhood socioeconomic context (NSEC) index score. The final NSEC index score was modeled by quartile. A higher score (and therefore higher quartile) represented more extreme positive scores on constituent factors, representing more adverse overall NSEC. | **HIV viral suppression:** Viral suppression was defined as an HIV-1 RNA <200 copies/mL.<br>More adverse socioeconomic context was significantly associated with lack of viral suppression for the 4th vs. the 1st NSEC quartile (RR = 0.88; 95% CI: 0.80–0.97). |

*(Continued)*

**Table 3.** (Continued)

| No. | Study | Population | Exposure: Neighborhood characteristics | HIV treatment outcomes |
|---|---|---|---|---|
| | Ridgway et al. 2018 | Men and women attending the University of Chicago HIV Clinic, United States | The crime rates were normalized using block group demographics from the US Census Bureau. Average crime rates within two blocks of each individual patient route were then calculated. While crime rates were not calculated for the entire study period, crime rates within the study area fluctuate yearly by 5% or less. | **HIV Viral suppression:** Defined as viral load ≤ 200 copies/ml at the most recent visit. There was no difference in violent crime rate along the travel route between client's retention in care versus clients not retained in care. Distance and travel time from the patients' home to clinic were not associated with viral suppression, but a trend towards lower violent crime rates along the travel route for clients who were virally suppressed versus those not virally suppressed was observed (0.12 violent crimes/100,000 population versus 0.13 violent crimes/100,000 population, p = 0.07). |
| | Jiang et al. 2021 | Men and women living with HIV, 18 years and older in a large immunology clinic in South Carolina, United States | Two indicators were used for neighborhood SES:<br>• percentage of residents with high school education<br>• median annual household income<br>A neighborhood SES index was calculated by summing z-scored composites of these two indicators, with a higher score reflecting higher neighborhood SES. | **ART adherence:** ART adherence was measured using three items. Participants were asked to report whether they missed any doses of ART adherence "yesterday," "the day before yesterday," and "last Saturday/ Sunday" on a binary response (yes/no). Participants' responses on the three items were dichotomized into two categories: 0 = suboptimal adherence (i.e. doses missed at least one of these days) and 1 = optimal adherence (i.e. no doses missed on any of these days).<br>There were no direct effects of neighborhood SES on ART adherence (β = 0.01, p = 0.91). In the mediation model, however, neighborhood SES was associated with adherence self-efficacy (β = 0.18, p < 0.001). Adherence self-efficacy, in turn, was associated with ART adherence (β = 0.35, p < 0.001). The association between neighborhood SES and ART adherence was not statistically significant (β = −0.07, p = 0.23). There was a significant indirect effect of neighborhood SES on ART adherence through adherence self-efficacy (z = 2.93, p = 0.003).<br>In the combined model, neighborhood SES remained significantly associated with adherence self-efficacy (β = 0.12, p = 0.025), which in turn was related to ART adherence (β = 0.33, p < 0.001), controlling for individual SES. Neighborhood SES remained not associated with ART adherence (β = −0.07, p = 0.27). The indirect effect of neighborhood SES on ART adherence through adherence self-efficacy remained significant (z = 2.03, p = 0.042). The results were similar after controlling for both individual SES and covariates, with an exception that the indirect effect of neighborhood SES on ART adherence became nonsignificant (z = 1.79, p = 0.074). The model accounted for 10 percent of the variances in adherence self-efficacy and 12 percent of the variances in ART adherence. |
| | Mauck et al. 2018 | Men and women, 13 years old and older in the Florida HIV surveillance registry between 2000–2014, United States | Seven neighborhood-level socioeconomic (SES) indicators included to construct a SES index:<br>• percent below poverty<br>• median household income<br>• percent of households with annual income <$15,000<br>• percent of households with annual income ≥$150,000<br>• income disparity<br>• percent of population age ≥25 with less than a 12th grade education<br>• high-class work<br>Standardized scores for the seven variables were added and the scores were categorized into quartiles. Additional indicators were used separately below:<br>• percent of households that are composed of male-male unmarried partners was greater than or equal to 1%, the neighborhood was classified as "gay" in this study. Otherwise, it was classified as "not gay."<br>• Rural/urban status of the ZIP code was based on categorization C of Version 2.0 Rural- Urban Categorization (RUCA) data codes, developed by the University of Washington.<br>• percentage of NHB population within each ZCTA was used to measure racial composition. The percent NHB population was grouped into three categories: <25%, 25–49%, and ≥50%. | **HIV viral suppression:** Viral suppression undefined<br>Gay neighborhood residence was a protective factor for viral suppression in the crude model 1.41 (1.29–1.54; <0.0001) but was not significant in the adjusted models (Table II, model 4) or when stratifying by race/ethnicity (Table IV).<br>A > 50% Percent population non-Hispanic Black was associated with not being viral suppression (PR 1.78; 95% CI 1.65–1.93; p<0.0001) among MSM diagnosed with HIV between 2000 and 2014 in Florida<br>When stratifying by race/ethnicity, rural compared to urban residence was a protective factor for viral suppression among NHBs (OR = 0.66; 95% CI 0.50–0.87; p = 0.0039) (Table IV) among those with mode of transmission listed as MSM or MSM/IDU diagnosed with HIV between 2000 and 2014 in Florida.<br>Those in the Lowest SES quartiles was associated with not being virally suppressed (PR 1.70; 95% CI 1.52–1.90; p<0.0001) among MSM diagnosed with HIV between 2000 and 2014 in Florida |
| | Chen et al. 2019 | Men and women, 18 years old and above attending the Outpatient clinic at McCord Hospital, Durban, South Africa | Automobile ownership, non-spouse family members paying for care and neighborhood-level poverty were treated as the SES exposure variables.<br>• Neighborhood-level poverty was summarized as percent of households in each Main Place living under the annual income of R9600 (~ $1448 USD as of 1/1/2011), which was the income bracket created by the 2011 Census that was closest to the upper national poverty line—the annual household income of R7400 (~ $1116 USD as of 1/1/2011) after adjusting for the March 2011 inflation rates.<br>A trichotomous neighborhood level poverty variable was created by using the tertile cutoffs of the neighborhood-level poverty percentage distribution (i.e., < 22.4%, 22.4–33.9%, > 33.9%) labeled as low-poverty, moderately poor, and severely poor neighborhoods, respectively. | **HIV Viral suppression:** Virologic Failure (VF) was defined as having a single viral load measurement of >1000 copies/mL within 1–2 weeks of a visit to the clinic.<br>Living in severely poor neighborhood remained associated with increased odds of VF among women after adjusting for individual-level SES and the Euclidean distance to McCord Hospital (AOR 2.14, 95% CI 1.06–4.30, p = 0.033).<br>Compared to women, men were more likely to have greater VF odds in low-poverty (AOR 3.24, 95% CI 1.35–7.75, p = 0.008) and moderately poor neighborhoods (AOR 2.39, 95% CI 1.11–5.12, p = 0.025); but not in severely poor neighborhoods (AOR 1.13, 95% CI 0.51–2.54, p = 0.761), after adjusting for individual-level SES and the Euclidean distance to healthcare<br>Automobile ownership and non-spouse family members paying for healthcare were independently associated with increased VF among men and women, respectively, in moderately poor (AOR _automobile ownership_ 7.30, 95% CI 1.73–30.80, p = 0.007; AOR _non-spouse family paying care_ 4.35, 95% CI 1.77–10.69, p = 0.001) and severely poor neighborhoods (AOR _automobile ownership_ 6.33, 95% CI 4.08–9.82, p < 0.001; AOR _non-spouse family paying care_ 4.50, 95% CI 2.42–8.33, p < 0.001). The associations between the gender-specific individual-level SES exposure and VF was attenuated in low-poverty neighborhoods |

(*Continued*)

**Table 3.** (Continued)

| No. | Study | Population | Exposure: Neighborhood characteristics | HIV treatment outcomes |
|---|---|---|---|---|
| | Gebregzi et al. 2019 | Male and female youth, 13–24 years old in the Florida HIV surveillance registry between 1993–2014, United states | We used principal component analyses (PCA) to create a SES index (categorized into quartiles). The final SES index included 7 indicators, namely,<br>• percent below federal poverty level,<br>• median household income,<br>• percent of households with annual income <$15,000, percent of households with annual income ≥$150,000,<br>• income disparity,<br>• percent of population age ≥25 with less than a 12th grade education, and<br>• high-class work (managerial, business, science, and arts occupations).<br>Additionally, NHB density (percentage of NHB from total population and a proxy for segregation) was categorized into three levels (<25%, 25–50% and ≥50%).<br>Rural-urban categorization was based on Categorization C Version 2.0 of Rural-Urban Commuting Area (RUCA) codes, developed by the University of Washington. | **HIV Viral suppression:** Viral suppression was defined as having evidence of a viral load <200 copies/mL. None of the neighborhood factors were associated with viral suppression.<br>Non-Hispanic black density was not associated with viral suppression.<br>Rural-urban status was not associated with viral suppression. |
| | Jefferson et al. 2019 | Men and women, 13 years old and above in the NYC HIV Surveillance, United States | **Neighborhood SES**<br>United Hospital Fund (UHF)-Level Predictors:<br>• Racial/ethnic composition—low (<5% of residents were Black), medium (5–29%), and high (>30%).<br>• Age composition—The percent of residents who were 21–54 years old<br>• Food access environment—living in a food desert area (> 0.5 mile from a large grocery store or supermarket).<br>• Residential stability—the percent of residents residing in the same home in the last year and the male to female sex ratio for 18–64-year-old non-institutionalized residents<br>• Affluence—percent of residents ≥25 years old who were college educated; the percent of high-income households (>400% of 2009 U.S, median household income); and the percent of expensive homes (> 400% of 2009 U.S. median home value).<br>• Economic disadvantage—UHF-level median income; the percentages of people aged >16 years in the civilian workforce who were unemployed; of individuals living at or below the federal poverty level (< 200% of federal poverty limit); of households that had received public assistance in the last 12 months; and of adults >25 years old who did not have a high school degree/GED.<br>• poor access to healthcare—percent of UHF residents without health insurance and the percent of UHF residents who reported an unmet need for health care in the last 12 months<br>**Neighborhood Disorder**<br>• Social disorder—to assess dimensions of social disorder we created measures of the number of businesses licensed to sell alcohol for off-premises consumption per square mile in 2009 and the number of residences that had once been occupied and now were empty per square mile in 2009<br>• Police stop and frisk rates—stops per 100,000 adult residents and were categorized as low (<4,000 stops per 100,000 adult residents), medium (4,000–<20,000), and high (>20,000). Stops without an arrest per 100,000 adult residents were categorized: low (<6,000 stops without arrest per 100,000 residents), medium (6,000–<22,000), and high (≥22,000). | **HIV viral suppression:** "virally suppressed" was defined as having ≤200 copies of HIV per mL of blood at any point within 12 months after their diagnosis and "durably virally suppressed" if, within 12 months after their diagnosis, they (a) had at least two suppressed (≤200 cc/mL) viral load tests that were at least 90 days apart from one another with no intervening unsuppressed (>200 cc/mL) viral load tests, and (b) had no unsuppressed viral load tests after they had achieved durable viral suppression<br>*Racial/ethnic composition*<br>In the bivariate analysis, heterosexuals living in UHFs where less than 5% of residents were Black were 21% more likely to be suppressed than their counterparts living in UHFs where more than 30% of residents were Black (URR = 1.21, p = 0.01). The magnitude of this relationship was slightly attenuated in the multivariable analysis and attained only borderline statistical significance (ARR = 1.13, p = 0.09).<br>In the bivariate analysis, heterosexuals living in UHFs where less than 5% of residents were Black were 36% more likely to be durably suppressed than their counterparts living in UHFs where more than 30% of residents were Black (URR = 1.36, p = 0.002). The magnitude of this relationship was attenuated in the multivariable analysis and was no longer statistically significant (ARR = 1.05, p = 0.65).<br>*Age composition*—In bivariate and multivariable models, % residents who were 21–54 years old was not significantly associated with being virally suppressed or durable viral suppression.<br>*Food access environment*<br>In the bivariate analysis, heterosexuals living in UHFs where less than 1% of residents were food distressed were 25% more likely to be suppressed than their counterparts living in UHFs where at least 5% of residents were food distressed (unadjusted relative risk [URR] = 1.25, p = 0.04). The magnitude of this relationship was slightly attenuated in the multivariable analysis and was borderline statistically significant (adjusted relative risk [ARR] = 1.18, p = 0.09).<br>In UHFs where <1% of residents were food distressed, heterosexuals were 70% more likely to be durably suppressed than their counterparts living in UHFs where at least 5% of residents were food distressed (ARR = 1.70, p = 0.01; URR = 1.48, p = 0.02). These models also suggest a possible dose-response relationship: heterosexuals living in UHFs where between 1 to <5% of residents were food distressed had a marginally significant greater likelihood of achieving durable suppression than their counterparts living in UHFs where at least 5% of residents were food distressed (ARR = 1.36, p = 0.07; URR = 1.35, p = 0.07).<br>*Residential stability*–In bivariate and multivariable models, residential stability was not significantly associated with being virally suppressed or durable viral suppression.<br>*Affluence*—In bivariate and multivariable models, affluence was not significantly associated with being virally suppressed or durable viral suppression.<br>*Economic disadvantage*—In bivariate and multivariable models, economic disadvantage was not significantly associated with being virally suppressed or durable viral suppression.<br>*Poor access to healthcare*—In bivariate and multivariable models, poor access to healthcare disadvantage was not significantly associated with being virally suppressed or durable viral suppression.<br>*Neighborhood Disorder*<br>In bivariate and multivariate models, alcohol outlet density and residential vacancy was not significantly associated with being virally suppressed or durable viral suppression.<br>In bivariate and multivariate models, compared to high (≥20,000) stops per 100,000, low (<4,000) and medium (4,000–<20,000) stops were not significantly associated being virally suppressed or durable viral suppression.<br>In bivariate and multivariate models, compared to high (≥22,000) stops without an arrest per 100,000, low (<6,000) and medium (6,000–<22,000) stops were not significantly associated being virally suppressed or durable viral suppression. |

(*Continued*)

**Table 3.** (Continued)

| No. | Study | Population | Exposure: Neighborhood characteristics | HIV treatment outcomes |
|---|---|---|---|---|
| | Chandran et al. 2020 | Women in the Women's Interagency HIV Study (WIHS), United States | We evaluated the following structural measures: census-tract level education, poverty, vacant housing, unemployment, household income, household crowding, female-headed household, lack of car ownership, owner-occupied housing, and residential stability<br>• Education was defined as the percent of adults ≥25 years of age with a high school education/ equivalent or greater<br>• Poverty was defined as the percent of households living below the federal poverty line<br>• Vacant housing was defined as the percent of housing units that were vacant in the census tract<br>• Unemployment was defined as the percent of individuals ≥ 16 years of age that were in the labor force and were unemployed<br>• Household income was defined as the median household income for the census tract<br>• Household crowding was defined as the percent of houses with > 1 person per room<br>• Female-headed household was defined as the percent of households with a female head with at least 1 child<br>• Lack of car ownership was defined as the percent of households without access to a vehicle.<br>• Owner-occupied housing was defined as the percent of owner-occupied housing units.<br>• Residential stability was defined as the percent of households that had lived in the same housing unit one year prior. | **ART Adherence:** Adherence was measured by a self-report question of how often an individual took their ART as prescribed over the past 6 months. ≥ 75% adherence was considered optimal for this study.<br>Neighborhoods with increased education (aOR 4.84, 95% CI 4.83, 4.84) and owner-occupied housing units (aOR 1.26, 95% CI 1.01, 1.56) were positively associated with ART adherence.<br>Neighborhoods with increased poverty (aOR 0.52, 95% CI 0.33, 0.79), unemployment (aOR 0.28, 95% CI 0.11, 0.71), and lack of car ownership (aOR: 0.85, 95% CI 0.84, 0.86) were inversely associated with ART adherence.<br>Vacant housing, household crowding, female-headed households, and residential stability were not associated with ART adherence.<br>**HIV Viral suppression:** Defined as HIV RNA not detected or below the lower limit of detection (20 copies/ mL)<br>10-unit increases in proportions of census tract residents with at least a high school level of education (adjusted odds ratio (aOR) 1.37, 95% confidence interval (CI): 1.08, 1.75) and housing units occupied by the owner (aOR 1.17, 95% CI 1.05, 1.31) were associated with increased odds of achieving viral suppression.<br>Increases in households living below the poverty line (aOR 0.72, 95% CI 0.60, 0.88), unemployment (aOR 0.63, 95% CI 0.44, 0.89), female-headed households (aOR: 0.84, 95% CI 0.71, 0.99), and lack of car ownership (aOR 0.86, 95% CI 0.77, 0.96) were inversely associated with viral suppression.<br>Vacant housing, household crowding, and residential stability were not associated with viral suppression. |
| | Rojas et al. 2021 | Male and Females, 15 years old and above in the Florida HIV surveillance registry from January to December 2017, United States | Social determinants of health were assessed through 5 determinants at the zip code level:<br>*Economic stability*—employment status and poverty (variables: households below poverty level, government assistance, and median household income)<br>*Education*—was comprised of three domains: HS graduation, enrollment in higher education, and language and literacy. The variables used to represent these domains were categorized into less than an HS education and HS graduation only (HS graduation domain), greater than HS education (enrollment in higher education domain), and language other than English spoken at home (language and literacy domain).<br>*Social and community context*—used the civic participation and incarceration domains and included the following variables: census return rates by ZIP codes for the civic participation domain, and incarceration measured by rate of total incarcerations per population in 2017 within each ZIP code for the incarceration domain<br>*Health and healthcare*—assessed through the health insurance rate variable.<br>*Neighborhood and built environment*—was assessed by the domains of violent crime and access to foods. The uniform crime rates definitions for violent crime included aggravated assault, murder, aggravated stalking, rape, negligent manslaughter, robbery, and police shooting.<br>*Access to foods*—was measured by the percent of households within half-a-mile from a supermarket within each ZIP code. | **Uncontrolled HIV or non-HIV viral suppression** was defined as being out of treatment (i.e., did not have viral load test or a CD4 count) or had a viral load greater than or equal to 200 copies/ml.<br>The PCA, which included all SDOH variables at the ZIP code level, resulted in three factors.<br>Factor 1 included the education, economic stability, and health and healthcare determinants of the SDOH.<br>Factor 1 values ranged from − 2.29 to 1.94, with the higher positive numbers representing the lowest socioeconomic status (SES), lowest education, and least access to health insurance.<br>Factor 2 included social and community context and neighborhood and built environment determinants.<br>Factor 2 values ranged from − 0.72 to 2.62, with the higher positive numbers representing the highest crime rates and lowest civic participation rates.<br>Factor 3 included education and neighborhood and built environment determinants. Factor 3 values ranged from − 2.41 to 2.12, with the higher positive numbers indicating the lowest proportion of people with access to foods.<br>Unadjusted estimates indicated that Factors 2 and 3 directly affected the number of PLWH with uncontrolled HIV . However, when accounting for confounders, these factors lost significance, and only Factor 1 had a statistically significant effect on the outcome.<br>The regressions suggest that a one unit increase in Factor 1 results in a 16% (1.16 (C.I. 1.07, 1.26) increase in PLWH with uncontrolled HIV while holding all other variables constant. Similarly, a 1%increase in males produces a 470% (5.70 (C.I. 1.91, 17.02)) increase in PLWH with uncontrolled HIV, and a 1% increase in PLWH aged 25–44 creates a 161% (2.61 (C.I. 1.03, 6.60)) increase in PLWH with uncontrolled HIV.<br>The regressions also revealed a significant interaction between Factor 1 and the percentage of Whites in MDC ZIP codes. This interaction term reveals that among those PLWH who are high in Factor 1, being White was associated with a 66% (0.34 (C.I. 0.20, 0.58)) reduction in the number of PLWH with uncontrolled HIV. In other words, people who are White and low SES are significantly less likely to have uncontrolled HIV than the other racial/ethnic groups that are low SES. |
| | Cope et al. 2020 | Women in the Women's Interagency HIV Study (WIHS) with a median age of 49.3 from multiple areas in the United States. | Census tract-level poverty was defined as the proportion of each participant's census tract living below the federal poverty line during the past 12 months.<br>• We classified neighborhood poverty into three categories: ≤ 20%, > 20–40%, or > 40–100% of the total population living below the poverty line and categorized as low, high, and extreme poverty, respectively. | **ART Initiation:** ART status (defined as self-reported use of ≥ 3 antiretroviral medications, one of which is a protease inhibitor, a non-nucleoside reverse transcriptase inhibitor, one of the nucleoside reverse transcriptase inhibitors abacavir or tenofovir, an integrase inhibitor, or an entry inhibitor<br>Although most HIV-seropositive women were receiving ART, the proportion decreased as neighborhood poverty increased [≤ 20% poverty: N = 460/514 (89%); > 20–40% poverty: N = 575/657 (88%); > 40–100% poverty: N = 236/292 (81%); p = 0.002].<br>Among 192 HIV-seropositive women not receiving ART, most had seen a healthcare provider in the previous 6 months; although similar across poverty levels, the proportion was highest in women living in areas of extreme poverty [≤ 20% poverty: N = 42/54 (78%); > 20–40% poverty: N = 65/82 (79%); > 40–100% poverty: N = 50/56 (89%); p = 0.2].<br>**HIV Viral suppression:** unsuppressed viral load (VL) defined as VL > 200 copies/mL.<br>The prevalence of unsuppressed VL was higher among HIV-seropositive women living in neighborhoods of extreme poverty in both unadjusted analyses (> 40–100% versus ≤ 20% PR, 1.79; CI, 1.30–2.48) and analyses adjusted for individual-level markers of HIV disease, socioeconomic status, and demographics (> 40–100% versus ≤ 20% adjusted PR [aPR], 1.42; CI, 1.04–1.92).<br>The relationship between unsuppressed VL and poverty was attenuated and less precise among women living in neighborhoods of moderate poverty in unadjusted (> 20–40% versus ≤ 20% PR, 1.27; CI, 0.94–1.71) and adjusted (> 20–40% versus ≤ 20% aPR, 1.12; CI, 0.85–1.48) analyses |

*(Continued)*

**Table 3.** (Continued)

| No. | Study | Population | Exposure: Neighborhood characteristics | HIV treatment outcomes |
|---|---|---|---|---|
| | Gebreegziabher et al. 2020 | Men and women in the Alameda County HIV Surveillance registry between 2011–2016, United States | • Percent poor was defined as the percent living under the federal poverty in each census tract. The variable percent poor was rescaled for a 1- unit increase to represent a 10-percentage point increase in the percent poor in the census to facilitate interpretation of a meaningful increase in poverty.<br>• We defined foreign-born status (FB) as a binary variable wherein foreign-born individuals were those born outside the US, including naturalized citizens. Only those born in US mainland were considered US-born. | **HIV Viral suppression:** Undetectable viral load a year after diagnosis (referred here as viral load status) was defined as having less than 75 copies/mL of HIV virus in the blood on the last test in the year after diagnosis. Neighborhood poverty did not modify the association between foreign-born status and viral load status (p = 0.12). There was no meaningful pattern for the change in relative risk of the association between foreign-born status and outcome with increasing census tract poverty. |
| | Momplaisir et al. 2020 | Pregnant women with HIV, aged 25–34 in the perinatal HIV Exposure Reporting (PHER) program in Philadelphia over an 11-year period (2005–2015), United States | **Census Tract** of residence at delivery was used as a proxy for neighborhood in this analysis. Neighborhood measures, included extreme poverty, educational attainment, crime rates, and social capital, were obtained at the census tract level.<br>**Socioeconomic Status:** Census tract–level extreme poverty and educational attainment were obtained from the American Community Survey 5-year estimates for 2005 through 2009 and 2010 through 2014.<br>• Extreme poverty was defined as the proportion of residents with a ratio of income to poverty being less than 50% of the federally defined poverty level (calculated by dividing a family's income by the family's poverty income threshold).<br>• Educational attainment was assessed as the percentage of adults 25 years or older with a bachelor's degree or higher.<br>**Neighborhood Crime Rates:** Yearly crime rates per 10000 persons for violent, drug, and prostitution crimes were calculated by dividing the total number of incidents by the census-tract. To operationalize the crime index, we first standardized each measure of crime to mean (SD) 0 (1). Next, we calculated overall crime as the sum of these 3 standardized terms for each census tract and dichotomized overall crime at the median.<br>**Social Capital:** Social capital empirical Bayes estimates were derived from the Southeastern Pennsylvania Household Health Survey. Social capital varied from 0 to 3, with higher scores indicating higher social capital, and was based on the responses to the following questions and statements:<br>• "how likely are people in your neighborhood willing to help others,"<br>• "most people in my neighborhood can be trusted," and<br>• "I feel that I belong and am a part of my neighborhood."<br>Births were matched to the closest preceding social capital | **HIV Viral suppression:** The main outcome was elevated HIV viral load of > 200 copies/mL at delivery. In Models 1 and 2 for Pearson $\chi^2$ tests, each neighborhood exposure is included separately and adjusted for confounders (ie, year of birth, maternal age, race, previous birth while living with HIV, prenatal diagnosis of HIV). Model 2 built on Model 1 and includes adjustment for potential mediators (ie, prenatal substance use and adequacy of prenatal care). All models adjusted for clustering at the census tract level and for clustering for mothers with multiple births<br>*Socioeconomic status*<br>Extreme poverty was not associated with elevated viral load in model 1 (p = 0.70) or model 2 (p = 0.48).<br>After adjusting for year of birth, maternal age, race/ethnicity, previous birth while living with HIV, and prenatal diagnosis of HIV in Model 1, neighborhood education became negatively associated with having an elevated viral load; the adjusted odds ratio (AOR) for having an elevated viral load was 0.70 (95% CI, 0.50–0.96) for women living in neighborhoods above vs below the median of neighborhood education.<br>*Neighborhood crime rates*<br>Women living in neighborhoods above the median in violent crimes, drug crimes, and prostitution crimes had higher odds of having an elevated viral load at delivery compared with women in neighborhoods with rates below the median.<br>Violent crimes with an AOR of 1.51 (95% CI: 1.10–2.07; p = 0.01) and prostitution crimes with an AOR of 1.46 (95% CI: 1.06–2.00; p = 0.02) remained statistically significant and positively associated with having a higher HIV viral load at delivery in model 2.<br>Higher drug crimes remained associated with higher odds of having an elevated viral load at delivery, but the was not statistically significant with an AOR of 1.32 (95% CI: 0.96–1.82; p = 0.08).<br>The crime index also remained statistically significantly associated with elevated viral load at delivery after full adjustment: the AOR was 1.44 (95% CI, 1.05–1.98; p = 0.02) for women in neighborhoods with higher aggregated crimes compared to women in neighborhoods with lower aggregated crimes.<br>*Social capital*<br>Social capital was not associated with elevated viral load at delivery in model 1 (p = 0.63) or model 2 (p = 0.72). |
| | Khazanchi et al. 2021 | Men and women, 19 years old and above in the Ryan White–funded University of Nebraska Medical Center HIV Clinic | The Area Deprivation Index (ADI) is a validated, summative index of neighborhood-level inequalities composed of 17 education, employment, housing quality, and poverty measures drawn from 2011–2015 US Census American Community Survey data. ADI values represent national percentile rankings of neighborhood disadvantage from 1 to 100, with higher numbers indicating greater neighborhood disadvantage.<br>Rural/urban status of the ZIP code was based on categorization C of Version 2.0 Rural- Urban Categorization (RUCA) data codes, developed by the University of Washington. | **HIV viral suppression:** viral suppression was defined as most recent HIV-1 RNA <200 copies/mL. Poverty (aOR, 0.47; P = .002), and neighborhood deprivation (aOR, 0.88 for 10-unit increase in ADI; P = .023) were inversely associated with viral suppression.<br>Rurality was not associated with viral suppression. |
| | Olatosi et al. 2020 | Men and women, median age 34.8 in the South Carolina HIV Surveillance registry between 2014–2017, United States | Neighborhood deprivation scores were determined using the 2013 Area Deprivation Index (ADI) state ranks. The 2013 ADI is a validated index-based measure which uses U.S.<br>• Census poverty,<br>• education,<br>• housing<br>• employment<br>For this study, SC ADI state ranking deciles were grouped into three groups: 1 (ADI ranks 1–3) for least disadvantaged, 2 (ADI ranks 4–7) moderate disadvantaged, and 3 (ADI ranks 8–10) most disadvantaged neighborhoods.<br>Rural/urban residence was determined using the Rural Urban Commuting Area (RUCA) codes crosswalk with the zip codes of the PLWH residence [21]. Small rural, large rural, and isolated areas were aggregated into rural areas. | **HIV Viral suppression:** The outcome variable is the proportion of person-time (days) spent living with VL > 1500 copies/ml.<br>Individuals living in the moderate socioeconomically disadvantaged neighborhoods spent the least time living with VL > 1500 copies/ml (mean 31 days).<br>We did not find a difference in median time living with VL > 1500 copies/ml between rural and urban residents (46 days). |

(*Continued*)

**Table 3.** (Continued)

| No. | Study | Population | Exposure: Neighborhood characteristics | HIV treatment outcomes |
|---|---|---|---|---|
| | Wiewel et al. 2021 | NYC residents interviewed for CSBS, newly diagnosed with HIV between 2006 and 2012, and ≥ 13 years of age at diagnosis. | **Perceived neighborhood social cohesion** was calculated from the degree of participant agreement with three statements in the local NYC CSBS interview about trust, closeness, and helpfulness between neighbors, using a scored Likert-type scale from 1 (strongly agree) to 4 (strongly disagree). Cohesion was classified as low for persons with scores of 9–12 and high for persons with scores of 3–8. These statements were as follows: • I live in a close-knit neighborhood. • People in my neighborhood can be trusted. • People in my neighborhood are willing to help each other. Calculations of individual/household poverty drew from US Census Bureau weighted average poverty thresholds for 2013 by size of family unit (U.S. Census Bureau, N.D.d). Neighborhood poverty statistics at the level of ZIP code tabulation area (ZCTA) were acquired from the 2011 American Community Survey 5-year estimates based on the 2007–2011 surveys (U.S. Census Bureau, N.D.a; U.S. Census Bureau, N.D.b) and linked to CSBS cases through the ZIP code of patient residence at diagnosis. | **HIV viral suppression:** Defined as ≤400 copies of HIV RNA/mL plasma. Around 65 percent (n = 60) achieved suppression within 12 months of diagnosis. Suppression at 12 months was lower among persons reporting low cohesion (60.4%) than high cohesion (70.5%), but this difference was not statistically significant (p = 0.31). Almost the entire population (95.7%) achieved viral suppression by 30 June 2014 (i.e. they had at least one suppressed VL; not shown). The median number of days to first suppression was greater among persons who perceived low cohesion (204 days, vs 168 days among high cohesion, p = 0.02). However, Kaplan–Meier survival curves indicated no difference in rates of suppression between persons perceiving low and high cohesion (p = 0.95). Cohesion was not associated with suppression in unadjusted or fully adjusted models (hazards ratio (95% confidence interval) for persons reporting high versus low cohesion, unadjusted and adjusted, respectively: 0.99 (0.64–1.51) and 0.79 (0.49–1.28)). We found modest differences in the time from HIV diagnosis to viral suppression between persons perceiving low versus high neighborhood social cohesion. These differences disappeared in survival analysis, both before and after controlling for individual characteristics and neighborhood poverty. Percent of population below federal poverty threshold in past 12 months was not associated with being Virologically suppressed within 12 months (p = 0.2558) or Median time from diagnosis to suppression (p = 0.0869) |
| | Caleb-Adepoju et al. 2021 | Minority women (Hispanic, African American, and Haitian) enrolled in the Miami-Dade County Ryan White Program in 2017, 18 years of age and above | Neighborhood environment was described as low socioeconomic status (SES) and residential instability/ homicide | **HIV viral suppression:** Defined as having <200 copies/mL using the last viral load test in 2017 Having a household income of <100% federal poverty level (FPL) vs. ≥200% FPL (aOR:0.54: 95% CI: 0.30–0.95) and living in a residentially unstable neighborhood (aOR: 0.77, 95% CI: 0.64–0.93) were significantly associated with lower odds of viral suppression. |
| | Dawit et al. 2021 | Male and female clients, 18 years of age and above, who received medical case management, medical care, and viral load laboratory services in the Miami-Dade County Ryan White Program in 2017. | Neighborhood-level variables were selected based on social disorganization theory, while average number of homicides during 2013–2017 was used to assess neighborhood disorder. Neighborhood indices were developed for 24 variables that were collected from ACS and were used as a proxy for social characteristics in prior literature. This created three indices which were categorized in to tertiles (low, moderate, and high). • The SES disadvantage index was composed of 12 variables, including public assistance, vehicle ownership, crowding, income disparity, education, occupation, and employment. • The residential instability index consisted of rented housing and mobility. • Racial/language homogeneity index consisted of percent non-Hispanic Black (NHB) and English language proficiency. | **HIV sustained viral suppression:** Defined as having <200 copies/mL in all viral load laboratory tests during 2017. In low SES disadvantaged neighborhoods, NHB had a lower odd of sustained viral suppression [adjusted odds ratio (aOR):0.39; 95% confidence interval (CI): 0.20–0.74] compared to NHW in low SES disadvantaged neighborhoods. Haitians had lower odds of sustained viral suppression (aOR: 0.42;95% CI: 0.18–0.97) in moderate SES disadvantaged neighborhoods when compared to NHW in moderate SES disadvantaged neighborhoods. Lower odds of sustained viral suppression were observed for NHB (aOR: 0.31; 95% CI: 0.15–0.65) in moderate residential instability neighborhoods compared to NHW in same neighborhoods. NHB exhibited lower odds of sustained viral suppression in neighborhoods with low and high racial/language homogeneity (aOR: 0.38; 95% CI: 0.16–0.88) and (aOR: 0.38; 95% CI: 0.19–0.75), respectively, when compared to NHW in the same neighborhoods. Haitians residing in neighborhoods with high racial/language homogeneity had lower odds of sustained viral suppression (aOR: 0.49; 95% CI: 0.38–0.75) when compared to NHW in the same neighborhoods. |
| | Lopez et al. 2022 | PLWH 13 years old and above residing in the St. Louis region treated at the Washington University Infectious Disease Clinic from 2009 to 2015 within the Ryan White Program. | Neighborhood-level socioeconomic status (nSES) was characterized nSES using U.S. Census data linked to the patient database by zip code. nSES was classified into tertiles with "high nSES" and "medium nSES", respectively, compared to the referent "low nSES". Zip code level nSES variables included: • percent below poverty, • percent unemployed, • percent with health insurance, • percent with food stamp/SNAP benefits, • percent with a college education, • percent living with a disability, • percent of households with no vehicle. We combined the categories of "high nSES" and "medium nSES" into a single "high nSES" category due to the consistency of results between the "high nSES" and "medium nSES" categories with the outcome of VS. Thus, we present results comparing this combined "high nSES" category to that of "low nSES". | **HIV viral suppression:** calculated average of <200 copies/mL over the course of all their visits within the index year of observation. Univariate analyses showed all the nSES variables, such that living in areas with low poverty (1.30, 1.09–1.54), low unemployment (1.44, 1.21–1.71), low disability (1.32, 1.11–1.57), low independent living difficulty (1.38, 1.17–1.64), high health insurance coverage (1.33, 1.12–1.59), and high educational attainment (1.25, 1.05–1.48), were independently associated with attainment of viral suppression. |

(*Continued*)

**Table 3.** (Continued)

| No. | Study | Population | Exposure: Neighborhood characteristics | HIV treatment outcomes |
|---|---|---|---|---|
| | Hovhannisyan et al. 2022 | The HIV Clinical Cohort of the National Institute of Infectious Diseases (INI), at the Oswaldo Cruz Foundation (FIOCRUZ) who initiated ART and resided in the metropolitan region of Rio de Janeiro, Brazil between January 1, 2000, and December 31, 2017. | The municipal human development index (MHDI) and social vulnerability index (SVI) are multidimensional scores of population-level development and socioeconomic indicators. The indices quantify neighborhood characteristics, such as education, unemployment, and poverty, MHDI allows for comparison of development between local and national contexts. It ranges between zero and one, with values closer to one signifying higher development. The Atlas of Human Development in Brazil (ADHB) MHDI classification was adopted: very low to low (MHDI of 0–0.599), medium (MHDI 0.600–0.699), high (MHDI of 0.700–0.799), and very high (MHDI of 0.800–1.0). Provided by the Atlas of Social Vulnerability (ASV), the SVI was created to complement the MHDI. The SVI is an index of 16 measures grouped into: human capital, urban infrastructure, and work and income. The index ranges between zero and one, with values closer to zero signifying the lowest social vulnerability. The ASV SVI classification was adopted: high–very high (SVI of 0.4–1.0), medium (SVI of 0.3–0.399), and low–very low (SVI of 0–0.299). | Our outcome of interest was viral suppression at six months (or the closest result within a window of 90 to 270 days) after ART initiation date. In the adjusted multilevel regression models that considered MHDI and SVI as covariates, neighborhood socioeconomic indicators again did not predict viral suppression. In the model with MHDI, participants residing in neighborhoods with medium (aOR 1.06, 95% CI 0.81–1.39) and very low to low (aOR 1.12, 95% CI 0.84–1.50) development did not have significantly different odds of viral suppression compared to participants in high-to-very high MHDI neighborhoods. With very low- low SVI as the reference group, residing in neighborhoods with medium (aOR 1.12, 95% CI 0.87–1.43) and high-to-very high (aOR 1.21, 95%CI 0.85–1.71) vulnerability did not have statistically different odds of viral suppression. |
| | Wright et al. 2022 | Individuals residing in South Florida between October 2019 and January 2020. Inclusion criteria: 1. Aged 18 years and older 2. English speaking 3. Black and or African American 4. Cis-gender female 5. Living with HIV 6. Owning a cell phone with text messaging and Internet capability 7. capable of understanding and completing the informed consent process and procedures | Neighborhood characteristics was measured using the American Community Survey (ACS) 2019 5-year estimates, the National Center for Charitable Statistics (NCCS) Data Archive and the National Crime Victimization Survey (NCVS). • ACS: Neighborhood level employment rate, median income, education. • NCCS: number of Christian religious institutions, number of low-income and subsidized rental housing in the neighborhood. • Women's responses to the NCVS were also used as a proxy for neighborhood crime given barriers to reporting and underreporting. This survey assesses experiences of 7 major types of crime victimization—assault (aggravated and simple), burglary, robbery, identity theft, motor vehicle theft, rape, and sexual assault. | **HIV Viral suppression:** A VL cutoff of less than 200 copies per millimeter and a cutoff of less than 20 was defined as undetectable. Higher crime was associated with higher VL log (b50.000591; P<0.05). Higher education was associated with lower VL log (b5–4.715; P<0.05) and higher likelihood of HIV viral suppression (b56.844; P<0.05) and undetectable VL (b56.814; P<0.05). Higher employment was associated with higher likelihood of undetectable VL (b50.0296; P = 0.05). Higher median income was associated with a lower likelihood of HIV viral suppression (b5–0.0253; P<0.05) and undetectable VL (b5–0.0224; P<0.05). Contrary to the direction for within neighborhood, neighboring higher median income was associated with higher likelihood of undetectable VL (b50.00324; P<0.01). |

composite measure, making it difficult to assess the impact of each factor individually [68, 74, 75, 85, 87, 101].

With reference to ART initiation, the two studies that used neighborhood-level percent of residents with education and median income as SES indicators presented contradicting results: Joy et al. found no association with late access to Highly Active Antiretroviral Therapy (HAART) [67], while Druyts et al. found an association among neighborhoods concentrated with Injection Drug Users (IDUs) who had a lower HAART initiation rate compared to neighborhoods concentrated with gay men [79]. In terms of ART adherence, one study found no direct effects of neighborhood-level education and income on ART adherence, but found a significant indirect effect on ART adherence through adherence self-efficacy [86]. On the other hand, another study found that neighborhoods with increased education and owner-occupied housing units were positively associated with ART adherence, while lack of car ownership was inversely associated with ART adherence [81].

Out of the fifteen studies assessing HIV viral suppression, seven found no association with neighborhood level income [70, 72, 78], affluence [80], residential stability [80, 81], health care access [70, 80], and food accessibility [88, 99]. Six studies found inverse associations with HIV viral suppression and neighborhood level income [95], education [81, 82, 100], lack of car ownership [77, 81, 100], residential instability [91], percent of female-headed households [81], and food inaccessibility [80]. A positive association with HIV viral suppression was observed among men who have sex with men (MSM) living in areas that were more concentrated with MSM-headed households [70], neighborhoods with low poverty, low unemployment, high insurance coverage, and high education attainment [93]. Lastly, Chen et al. found non-spouse family members paying for healthcare was independently associated with increased virologic failure in moderately poor and severely poor neighborhoods in Durban, South Africa [77].

**Neighborhood deprivation.** This construct focuses specifically on the lack or insufficiency of resources. Of the 17 studies that assessed indicators of neighborhood deprivation, 3 examined outcomes on ART initiation [67, 79, 96], one study focused on ART adherence [81], six on HIV viral suppression [69, 70, 80, 81, 96, 100], and 10 studies partly or fully combined their neighborhood deprivation indicators into a composite measure [70, 72, 74, 75, 78, 80, 85, 87, 94, 97, 101]. All three studies found an inverse relationship between percent unemployment and ART initiation [67, 79, 96]. Chandran et al. [81] is the only study reporting findings on the association between neighborhood deprivation and ART adherence. It found percent unemployment to be inversely associated with ART adherence but found percent vacant housing not to be associated with ART adherence. Three studies found no association between HIV viral suppression and percent unemployment [96] or residential vacancy [80, 81]. However, three studies found inverse relationships between HIV viral suppression and percent unemployment [69, 81] and residential vacancy [100].

**Neighborhood disadvantage.** As seen in **Table 3**, among the 25 studies using the percentage of households or residents living below the poverty line as a neighborhood characteristic, four studies assessed ART initiation [67, 79, 83, 96], two studies assessed ART adherence [81, 98], and 13 studies assessed HIV viral suppression [69, 73, 77, 81–84, 87, 89, 96, 99, 100]. Nine studies combined poverty indicators with other characteristics to create a composite measure, making it difficult to assess the impact of living below the poverty line alone [70, 72, 74, 75, 78, 80, 85, 88, 101]. Six studies found no association between the percentage of households or residents living below the poverty line and ART initiation [67, 96] or HIV viral suppression [82, 84, 89, 96, 99]. However, 12 studies found an association between the percentage of households or residents living below the poverty line and ART initiation [79, 83], ART adherence [90, 98] and HIV viral suppression [69, 73, 77, 81, 83, 87, 91, 100]. ART initiation among PLWH decreased as the percentage of households or residents living below the poverty line increased

[83]. ART adherence similarly decreased among PLWH as the percentage of households or residents living below the poverty line increased [90, 98]. HIV viral suppression was also inversely associated with the percentage of households or residents living below the poverty line [69, 73, 77, 81, 83, 87, 91, 100] with the addition of gender and racial/ethnic differences [73].

**Neighborhood composition and interaction.** This construct focuses on residents' differential experiences and feelings of belonging. Among the 18 studies that assessed the composition and interaction in neighborhoods, four assessed its effect on ART initiation [67, 71, 79, 96], none on ART adherence, 13 on HIV viral suppression [69–72, 74, 78, 80, 82, 88, 89, 92, 96, 99], and three studies combined composition and interaction factors into a composite measure, making it difficult to examine the impact of each characteristic individually [68, 75, 87]. Three of the four studies on the associations with ART initiation found no association with Aboriginal residential density, [67] African American residential density [96], social cohesion, or social support [71]. Only Druyts et al. found a significant association between ART initiation and the percent of aboriginals residing in their neighborhood among neighborhoods concentrated with IDUs who had a lower HAART initiation rate compared to neighborhoods concentrated with gay men [79]. Nine out of the thirteen studies found no association between HIV viral suppression and African American residential density [72, 78, 96, 99], social capital [71, 82], social cohesion [71, 89], social support [71], and civic participation [88]. However, the remaining five studies found an inverse relationship between neighborhoods with higher percentages of African American residential density and HIV viral suppression [69, 70, 74, 80, 92].

**Urban/rural status.** This construct focuses on the level of urbanization and overall population densities. Seven studies assessed urban/rural status as a neighborhood characteristic [67, 72, 74, 78, 85, 87, 90]. In terms of ART initiation, Joy et al. found no association between rural-urban status and late access to HAART [67], while Ohl et al. found rural residents were somewhat more likely to initiate combination ART [90]. Three studies found no association between rural-urban status and HIV viral suppression [78, 85, 87], while two found associations [72, 74]. Sheehan et al. found that rural residence was associated with increased viral suppression for non-Latino Blacks and non-Latino Whites [72]. Likewise, Mauck et al. found that when stratifying by race/ethnicity, rural residence was associated with increased odds of viral suppression among Non-Hispanic Blacks with mode of transmission listed as MSM or MSM/IDU [74].

**Neighborhood disorder.** This construct focuses on safety, and physical and social order. Among the 11 studies that assessed components of neighborhood disorder, one study assessed the effects of neighborhood disorder on ART initiation [98], 2 studies assessed the effects of disorder on ART adherence [71, 95], and three studies assessed the effects of disorder on HIV viral suppression [70, 76, 80, 82, 88, 91, 92, 95, 99]. Shacham et al. found that greater levels of perceived fear were associated with a lower likelihood of having a current combination antiretroviral therapy (cART) prescription [71] while Wright et al. found higher crime was associated with lower adherence [95]. Likewise, Surratt et al., the only study reporting ART adherence outcomes in relation to disorder, found that participants reporting higher neighborhood disorder had higher odds of diversion-related nonadherence [98]. Among the six studies that reported HIV viral suppression outcomes, three studies found no association between HIV viral suppression and poor-quality environment [99], stop and frisk rates [80], crime [88, 95], and civic incarceration rates [88]. However, three studies found inverse relationships between HIV viral suppression with stop and frisk rates among specific racial groups [70] and crime [76, 82].

## Discussion

Neighborhood characteristics and HIV treatment outcomes research has burgeoned in the last decade. There is also a global disparity in this research area as 97% of studies were conducted in high-income countries. Most of the studies in this review characterized neighborhoods with socio-economic factors that describe resources individuals in each neighborhood have that may impact their accessibility of healthcare and its influence on health behavior within the community. Although findings were mixed, the majority of studies in this review found that HIV treatment outcomes were inversely associated with diminished neighborhood-level socio-economic factors. Fewer studies in this review characterized neighborhoods with social factors that contribute to health behavior through pathways of culture and social norms (e.g., demographic composition, social interactions, and support within the neighborhood). Some studies in this review found racial/ethnic group residential density, neighborhood disorder, perceived fear, policing, and crime to be inversely associated with HIV treatment outcomes. In terms of rural-urban status, although most found null effects, some found protective effects of rural residence on HIV treatment outcomes.

Addressing healthcare disparities related to SES at the community level requires efforts to improve affordability, accessibility, and quality of healthcare services. This includes addressing indicators such as median household income, education, poverty rates, proximity to healthcare facilities, transportation infrastructure, employment, health insurance coverage, etc. SES factors have been associated with daily investments into the production of health: over time, individuals with lower SES have deteriorated health compared to those with higher SES due to the resources available to them to access health-promoting goods and services [102]. Most of the SES factors are intertwined. For example, education improves healthcare utilization through many pathways such as employment opportunities that offer more income which then increases assets and access (e.g., housing, cars, health insurance, food, etc.) that facilitate promotive health behavior [103]. At the Neighborhood-level, education may also indicate collective human capital with different skill sets that can come together to improve availability of health-promoting goods and services in their neighborhood (e.g., bike lanes, sidewalks, recreational activities, etc.). Higher SES neighborhoods may also collectively advocate for better food access and municipal resources [104]. Home-ownership may indicate residents' collective interest in preserving the quality and safety of a neighborhood for property value purposes [105]. Differences in neighborhood SES levels also extend to differences in healthcare quality: lower quality in lower SES than higher SES communities can be ascribed to fewer resources, scarcity of providers, and more healthcare needs due to syndemics of poverty [106, 107]. Poverty itself has implications for healthcare access and utilization. After paying for basic necessities, individuals living more than 50% below the poverty line also must consider their health and medical visit needs with very few resources remaining [108]. Healthcare for PLWH with low income is usually covered by government programs, non-governmental assistance programs, private insurance, or a combination of these [109]. Although largely beneficial, some of the programs come with various complex eligibility criteria and enrollment processes which can be a barrier. Transportation also affects access and utilization of healthcare, especially for those with lower income or who are under/uninsured [110]. Transportation vulnerability can arise from insufficient transportation infrastructure in communities and inharmonious fit between transportation and health systems [111]. A consequence of transportation vulnerability is reliance on putting together multiple modes of transport to access care and ancillary services, creating additional barriers to service utilization and medical adherence [111].

In terms of community-level social resources, the use of the racial/ethnic density effect is born out of the theory that suggests racial/ethnic groups living in areas concentrated with

people of the same racial/ethnic group are healthier than those in areas of less concentration [46]. Higher racial/ethnic density fosters stronger social networks and support systems, ideally providing access to cultural appropriate health resources that enhance health literacy and improve health outcomes [46]. However, racial/ethnic density can also contribute to experiences of discrimination and minority stress, which negatively impact health and well-being [112]. Our findings were consistent with a review on the effects of ethnic density on morbidity, mortality, and health behaviors, in which a majority of the studies had null findings. However, those with associations found detrimental associations with health in areas with higher African American density [113]. The wide usage and generalizations about the effects of racial/ethnic density under this theory has been cautioned, given the documented heterogeneity in immigrant history and status [114, 115]. Differences in African American density effects have been found in subgroup analysis patterned by nativity, age, and gender [113]. The concept of density can be applied to LGBTQ+ communities, suggesting that higher density can foster stronger social networks and support systems, leading to improved health outcomes. However, it is crucial to recognize the unique challenges faced by LGBTQ+ individuals, including discrimination and stigma, which can impact their health and well-being. Therefore, having more MSMs in one's community could translate to more social support for their unique circumstances when it comes to their HIV care. In a study among MSM, HIV-related social support with a peer component was a facilitator of optimal testing and awareness of novel prevention strategies (e.g., self-testing or PrEP) [116, 117]. Crime and policing activities have an impact on utilization of care, and thus on HIV care management, although no direct links were mentioned in the studies in this review. Neighborhood quality perceptions have been linked to utilization of care, as a study using data from the Dallas Heart Study found that individuals with a more unfavorable perception of their physical environment were significantly more likely to report a lack of a usual source of care and longer time periods since their last routine visit [118]. Also, individuals with a more unfavorable perception of greater neighborhood violence reported longer time periods since their last routine visit [118].

Rural areas are usually cited to have lower access to health information from specialist doctors than urban residents, lower health care coverage, increased travel times to specialist doctors than urban areas, and lack of access to transportation among rural residents compared to urban residents [119]. Therefore, it is surprising to find rural residence as protective for HIV treatment outcomes. HIV-related disparities continue to persist more in rural than urban areas, due to stigma and enhanced risk associated with sexual orientation, particularly in communities of color, especially since high rates of poverty, racism, drug use, and poor access to care may be more profound in rural areas than in urban [112]. On the other hand, rural areas have higher social capital and social cohesion within the rural community [120, 121] which my help buffer the socio-economic stressors [122].

## Strengths and limitations

This review had some strengths and limitations. Twenty-nine out of thirty-five studies were conducted in high-income countries and only one in middle- or low-income countries. This discrepancy may limit the generalizability of findings, especially since lower income countries bear the brunt of the HIV epidemic. However, the fact that we identified 35 studies that include various populations considered high-risk groups (e.g., sexual minorities, racial/ethnic minorities, IDUs, and women) in addition to the general population of PLWH is a strength. As our knowledge and innovation in HIV research grew over time, definitions for HIV treatment outcomes changed, which also affected the synthesis of the findings. For example, the definition of HIV viral suppression has changed over time; thus, this review included studies

that used various cut-off points for viral suppression ($\leq$ 1000 copies/ml, <400 copies/ml $\leq$ 200 copies/ml). However, this is a testament to the innovation in testing technology. The review also covered multiple definitions of geographical, social, and economic neighborhood characteristics, which was a strength as well as a limitation. The use of composite scores to form SES or deprivation indexes in some studies made it difficult to look at some neighborhood characteristics independently and compare studies. In addition, among those that used composite scores, many did not use the same indicators nor the same number of indicators, limiting possible comparisons. Lastly, the scoping review methodology although systematic in approach, aims to provide a wide-ranging overview of a topic rather than a critical analysis, thus does not assess quality of included studies reducing its rigor compared to systematic reviews.

## Public health implications

Our findings identified some areas for further neighborhood and HIV research. There continues to be a need to develop theories and more specific hypotheses on the dynamic processes through which neighborhood and individual factors may together impact specific health outcomes in order to inform future interventions [23]. Research suggests that there are four mechanisms through which neighborhood characteristics may affect an individual's ability to access care, including neighborhood-level information networks, health behavior norms, social capital, and healthcare resources [24]. Social capital and healthcare resources significantly predict an individual's care access, and since differences in care access may explain individual-level health disparities between neighborhoods, policies designed to improve care access must account for both individual and neighborhood factors [24]. Our review findings suggest a clear focus on epidemiological research rather than implementation or interdisciplinary research. Future studies, in addition to assessing the epidemiology, should be designed to target specific policies or neighborhood resources so that they can be translated into interventions. For example, studying the effect of rural-urban status without identifying the differences in available resources (e.g., HIV specialist care, ancillary resources, public transportation, distance to services, HIV advocacy organizations, etc.) between the two may elucidate possible reasons for mixed findings. A recent review highlighted the geospatial variation in ART success and inequitable distribution of HIV care in racially segregated, economically disadvantaged, and increasingly rural areas [123]. It also highlighted the utility and limitations of using GIS to monitor health and HIV treatment outcomes and the need for careful planning of resources with respect to the geospatial movement and location of PLWH [123]. Many studies in this review in part or wholly used composite scores to describe neighborhood characteristics. Although they are convenient and a popular method of capturing complex or multidimensional concepts at the whole population level using a wide variety of data sources (e.g., census, administrative, and geographical) and methods, researchers do not often justify their decision with respect to the data or methods used to develop a small-area measure; future studies should include justifications [124].

## Conclusion

Neighborhood and HIV treatment outcomes research has flourished in the last few years, providing valuable insight into various social and socio-economic neighborhood factors that may be associated with HIV treatment outcomes among PLWH. Overall, the findings were mixed on the impact of neighborhood characteristics on HIV treatment outcomes. However, it's clear that understanding community level socio-economic resources and social support capacity are essential to improving health outcomes among PLWH. In addition, identifying

standardized and consistent ways in which to describe neighborhoods. Future research in this topic should include more interdisciplinary approaches that may aid in elucidating pathways of association. Moreover, attaching neighborhood resource availability to the research may also facilitate the translation of findings to interventions.

## Supporting information

**S1 File. The conceptual model for this review adapted from the Social Ecological Model.** (TIF)

**S2 File. Lkimaru_PRISMA ScR checklist.** (PDF)

## Acknowledgments

I thank Jean McClelland (JM) for their expertise as a Medical Librarian and assistance throughout all aspects of this scoping review.

## Author Contributions

**Conceptualization:** Linda Jepkoech Kimaru.

**Methodology:** Linda Jepkoech Kimaru.

**Writing – original draft:** Linda Jepkoech Kimaru.

**Writing – review & editing:** Magdiel A. Habila, Namoonga M. Mantina, Purnima Madhivanan, Elizabeth Connick, Kacey Ernst, John Ehiri.

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
