## [Decision Letter · Decision Letter 0]

16 May 2023

PGPH-D-23-00460

Neighborhood characteristics and HIV treatment outcomes: A scoping review

Dear Dr. Kimaru,

Thank you for submitting your manuscript to PLOS Global Public Health. After careful consideration, we feel that it has merit but does not fully meet PLOS Global Public Health’s publication criteria as it currently stands. Therefore, we invite you to submit a revised version of the manuscript that addresses the points raised during the review process.

The manuscript was evaluated by two reviewers who provided different opinions. Reviewer 1 raised a couple of concerns that need to be addressed.

We look forward to receiving your revised manuscript.

Kind regards,

Jianhong Zhou

Staff Editor

Journal Requirements:

1. Please ensure that you have addressed all items recommended in the PRISMA checklist and include the PRISMA flowchart as Figure 1.

2. Please send a completed 'Competing Interests' statement, including any COIs declared by your co-authors. If you have no competing interests to declare, please state "The authors have declared that no competing interests exist". Otherwise please declare all competing interests beginning with the statement "I have read the journal's policy and the authors of this manuscript have the following competing interests:"

3. Please amend your online Financial Disclosure statement. If you did not receive any funding for this study, please simply state: “The authors received no specific funding for this work.”

Additional Editor Comments (if provided):

Reviewers' comments:

Reviewer's Responses to Questions

**Comments to the Author**

1. Does this manuscript meet PLOS Global Public Health’s publication criteria? Is the manuscript technically sound, and do the data support the conclusions? The manuscript must describe methodologically and ethically rigorous research with conclusions that are appropriately drawn based on the data presented.

Reviewer #1: Partly

Reviewer #2: Yes

2. Has the statistical analysis been performed appropriately and rigorously?

Reviewer #1: Yes

Reviewer #2: Yes

3. Have the authors made all data underlying the findings in their manuscript fully available (please refer to the Data Availability Statement at the start of the manuscript PDF file)?

Reviewer #1: Yes

Reviewer #2: Yes

4. Is the manuscript presented in an intelligible fashion and written in standard English?

Reviewer #1: Yes

Reviewer #2: Yes

5. Review Comments to the Author

Reviewer #1: My comments are as follows:

Abstract:

- The first sentence is very dismissive of people living with HIV. In this era, people are living with HIV for many years and there is much literature detailing the unique health-related issues PLWH face. I recommend the first sentence be appropriately revised so as not to turn off readers right at the outset.

- The justification for why this paper is important is not compelling in the abstract – I recommend a more honed introduction section.

Background:

- The first few paragraphs read as if they was written decades ago. The benefits of ART and the HIV care cascade are well-established, and I encourage the authors to quickly get to the issue of neighborhood factors given this is the focus of their paper.

- I would not focus so much on where neighborhood-level data is obtained, but instead delve more deeply into issues of why neighborhood characteristics are important (separate from individual-level factors), and opportunities they bring for intervention.

Methods:

- There is no justification why particular neighborhood-level indicators were deemed as important by the authors. For example, “physical decay” was included, but not “residential stability”. The choices they made are fine, but need to be justified.

Results:

- The constructs of “SES” or “Neighborhood Deprivation” are likely treated differently in each study given the lack of clarity in definition of these constructs. This needs to be better differentiated in the text.

- There is discussion of “racial/ethnic group density” that is not nuanced or carefully interpreted. Given the discourse surrounding how race and ethnicity should now be talked about, this text needs to be better developed.

Discussion:

- As indicated above, constructs such as SES and “race/ethnicity densities” are taken as a given, without a nuanced discussion of what these mean. This needs to be improved.

- Several sweeping sentences are delivered (i.e., Lines 317/318) that make generalizations regarding particular population subgroups without references or careful theory-driven discussions.

- The conclusions go beyond the scope of the review, talking about mechanisms, etc. I would focus the end to what the analysis actually found.

Reviewer #2: This paper presents the results of a model-based scoping review of the evidence regarding the association between neighborhood characteristics and HIV treatment outcomes. The methods and analysis are appropriate for the aims of the study and the discussion and implications are thorough. Altogether, the paper is well-written, informative, and contributes to the future direction of the research.

6. PLOS authors have the option to publish the peer review history of their article (what does this mean?). If published, this will include your full peer review and any attached files.

**Do you want your identity to be public for this peer review?** For information about this choice, including consent withdrawal, please see our Privacy Policy.

Reviewer #1: No

Reviewer #2: **Yes: **Jane Burke-Miller

---

## [Decision Letter · Decision Letter 1]

27 Jul 2023

PGPH-D-23-00460R1

Neighborhood characteristics and HIV treatment outcomes: A scoping review

Dear Dr. Kimaru,

Thank you for submitting your manuscript to PLOS Global Public Health. After careful consideration, we feel that it has merit but does not fully meet PLOS Global Public Health’s publication criteria as it currently stands. Therefore, we invite you to submit a revised version of the manuscript that addresses the points raised during the review process.

We look forward to receiving your revised manuscript.

Kind regards,

Hien Thi HO, PhD, MD

Academic Editor

Journal Requirements:

1. Please ensure that you have addressed all items recommended in the PRISMA checklist and include the PRISMA flowchart as Figure 1.

Additional Editor Comments (if provided):

**I saw that the authors have made efforts to address comments from reviewers. However, there are still areas that require attention. I think this manuscript needs much more work to be published in PGPH.**

**The authors should address the following issues to make the paper clearer and meet the publication criteria of a scoping review and PGPH journal.**

**The scoping review focused on the association of neighborhood characteristics and HIV treatment outcomes. However, in the introduction, the manuscript focuses on the lack of information on community factors and its relationship with HIV treatment outcomes. Then the manuscript talks about neighborhood characteristics. It is important to show how neighborhood characteristics links with community factors (through studies/) and what are neighborhood characteristics are (with definitions and examples, and references).**

**There are lack of definitions and references for key concepts in the manuscript. For example, the definition of HIV outcomes and neighbour characteristics need to be provided with references. How HIV outcomes are measured? References are needed on neighbourhood characteristics definition, similarly, citation is needed: line 93-95 “The literature describes neighborhood characteristics in many ways using the tools, indicators, and datasets to characterize a neighborhood’s socioeconomic status (SES), deprivation index, disorder status, economic disadvantage, and overall environment”**

**Importantly, there is a lack of justification why scoping review approach is used to address the review questions. The authors decided to use scoping review approach to address the review questions on assessing the influence of the association (Line 25) or examine the association (Line 89). Please use the terms and objectives consistently throughout the manuscripts.  As the authors use Arskey and O'Malley framework, how this framework could help to do review on the impact/ association of neighborhood characteristics and HIV treatment outcomes. It is important that the review specifies why scoping review is an appropriate approach, particularly why and how it can “critically examine the associations” (line 103) as no assessment of methodological quality nor formal data systhesis undertaken. Please clarify.**

**In the introduction: further information/ with examples is needed on how community factors links with neighbour characteristics. Please be consistence with influence and association in the introduction. Data on 2022 on HIV should be updated.**

**It is recommended that the authors should present clearly in the manuscript whether there are similar reviews on the topics.**

**How neighbourhood characteriscs were extracted and why should be further detailed.**

**Search strategy:**

**There is no limit on date of search, however, as HIV has been around since 1990s. Furthermore, the studies included in the scoping review only from 2006. Therefore, lower date limit for searches should be reconsidered.**

**Line 109: Databases’s platforms should be provided (Ovid, for instance).**

**Implications for research and limitations (on using scoping review to address the review question on associations) should be provided.**

Reviewers' comments:

Reviewer's Responses to Questions

**Comments to the Author**

1. If the authors have adequately addressed your comments raised in a previous round of review and you feel that this manuscript is now acceptable for publication, you may indicate that here to bypass the “Comments to the Author” section, enter your conflict of interest statement in the “Confidential to Editor” section, and submit your "Accept" recommendation.

Reviewer #1: All comments have been addressed

Reviewer #2: All comments have been addressed

2. Does this manuscript meet PLOS Global Public Health’s publication criteria? Is the manuscript technically sound, and do the data support the conclusions? The manuscript must describe methodologically and ethically rigorous research with conclusions that are appropriately drawn based on the data presented.

Reviewer #1: Yes

Reviewer #2: Yes

3. Has the statistical analysis been performed appropriately and rigorously?

Reviewer #1: N/A

Reviewer #2: N/A

4. Have the authors made all data underlying the findings in their manuscript fully available (please refer to the Data Availability Statement at the start of the manuscript PDF file)?

Reviewer #1: Yes

Reviewer #2: Yes

5. Is the manuscript presented in an intelligible fashion and written in standard English?

Reviewer #1: Yes

Reviewer #2: Yes

6. Review Comments to the Author

Reviewer #1: I appreciate the changes made the manuscript based on prior comments. I have no further comments or suggetsions.

Reviewer #2: This scoping review is well-researched and clearly presented. The results and discussion are a useful contribution to the study of neighborhood contexts and HIV outcomes.

7. PLOS authors have the option to publish the peer review history of their article (what does this mean?). If published, this will include your full peer review and any attached files.

**Do you want your identity to be public for this peer review?** For information about this choice, including consent withdrawal, please see our Privacy Policy.

Reviewer #1: No

Reviewer #2: No

---

## [Editor Report · Decision Letter 2]

10 Oct 2023

PGPH-D-23-00460R2

Neighborhood characteristics and HIV treatment outcomes: A scoping review

Dear Dr. Kimaru,

Thank you for submitting your manuscript to PLOS Global Public Health. After careful consideration, we feel that it has merit but does not fully meet PLOS Global Public Health’s publication criteria as it currently stands. Therefore, we invite you to submit a revised version of the manuscript that addresses the points raised during the review process.

We look forward to receiving your revised manuscript.

Kind regards,

Hien Thi HO, PhD, MD

Academic Editor
---

## [Editor Report · Decision Letter 3]

11 Jan 2024

Neighborhood characteristics and HIV treatment outcomes: A scoping review

PGPH-D-23-00460R3

Dear Ms. Kimaru,

We are pleased to inform you that your manuscript 'Neighborhood characteristics and HIV treatment outcomes: A scoping review' has been provisionally accepted for publication in PLOS Global Public Health.

Best regards,

Hien Thi HO, PhD, MD

Academic Editor